# Imputation of plasma lipid species to facilitate integration of lipidomic datasets

Aleksandar Dakic [1], Jingqin Wu [1], Tingting Wang[1], Kevin Huynh [1,2,3], Natalie Mellett[1], Thy Duong [1], Habtamu B. Beyene [1,2], Dianna J. Magliano [1], Jonathan E. Shaw [1], Melinda J. Carrington[1,3], Michael Inouye[1], Jean Y. Yang [4,5], Gemma A. Figtree[6,7], Joanne E. Curran [8], John Blangero [8], John Simes[9], LIPID Study Investigators*, Corey Giles [1,2,3,11] ✉ & Peter J. Meikle [1,2,3,10,11] ✉

Recent advancements in plasma lipidomic profiling methodology have significantly increased specificity and accuracy of lipid measurements. This evolution, driven by improved chromatographic and mass spectrometric resolution of newer platforms, has made it challenging to align datasets created at different times, or on different platforms. Here we present a framework for harmonising such plasma lipidomic datasets with different levels of granularity in their lipid measurements. Our method utilises elastic-net prediction models, constructed from high-resolution lipidomics reference datasets, to predict unmeasured lipid species in lower-resolution studies. The approach involves (1) constructing composite lipid measures in the reference dataset that map to less resolved lipids in the target dataset, (2) addressing discrepancies between aligned lipid species, (3) generating prediction models, (4) assessing their transferability into the targe dataset, and (5) evaluating their prediction accuracy. To demonstrate our approach, we used the AusDiab population-based cohort (747 lipid species) as the reference to impute unmeasured lipid species into the LIPID study (342 lipid species). Furthermore, we compared measured and imputed lipids in terms of parameter estimation and predictive performance, and validated imputations in an independent study. Our method for harmonising plasma lipidomic datasets will facilitate model validation and data integration efforts.

Plasma lipidomic profiling has become increasingly prevalent over the last decade, providing insights into previously unrecognised biological phenomena. Along with this rise in published work, the technology and methodology to perform plasma lipidomics has evolved significantly. The number, specificity, and accuracy of lipid measurements have increased greatly, along with throughput and robustness[1–4].

Current lipidomic assays are able to measure a large number of lipids (over 700) from a diverse array of classes (40 or more). Prior assays generally measured a smaller panel (about 300) of lipids, from a more selective number of classes. Despite the apparent similarities between platforms, it is often a challenge to closely align measurements. Due to improved chromatographic separation or increased mass spectrometer resolution, the level of granularity in newer platforms is often much higher. This means that older measurements can be composite measures of multiple lipid species that are entirely resolved in modern platforms. Using datasets with different numbers of lipid species and different levels of granularity presents a particular

A full list of affiliations appears at the end of the paper.  *A list of authors and their affiliations appears at the end of the paper.
✉e-mail: corey.giles@baker.edu.au; peter.meikle@baker.edu.au

challenge when building risk prediction models and validating them in different studies or when integrating data across population-based cohorts. Analysts are faced with a difficult choice between building models with a reduced number of lipid species common to all datasets or discarding older datasets and accepting reduced power and validation.

Here, we propose a framework for harmonising plasma lipidomic datasets produced with different levels of chromatographic resolution and mass accuracy. The principle behind our approach is a prediction (imputation) of the concentration of unmeasured lipid species in the target dataset, using a reference dataset. This approach relies primarily on the stable correlation structure across plasma lipidomic profiles, found both within and between lipid classes[3,5]. This allows us to build accurate predictive models for individual lipid species based on the more detailed lipidomic profiles in the reference dataset, and use them to predict the concentration of unmeasured lipid species in less detailed target dataset.

To demonstrate our approach, we used a large reference dataset, the AusDiab cohort[6] ($n = 10,339$), profiled with a contemporary, comprehensive lipidomic platform (747 lipid species) to impute unmeasured lipid species in an older, less comprehensively profiled study, the LIPID trial[7,8] ($n = 5991$, two timepoints; 342 lipid species). We further validated lipid imputations in another comprehensively profiled and ethnically distinct cohort, the San Antonio Family Heart Study (SAFHS)[9] ($n = 2595$, with 5590 complete observations of 795 lipid species across different phases of data collection).

## Results

### Lipidomic analysis of the AusDiab, LIPID and SAFHS studies
Our reference dataset, the AusDiab study, is a population-based cross-sectional survey and a prospective cohort study aimed at estimating the prevalence of diabetes mellitus, and its associated risk factors in the Australian adult population. The LIPID study is a randomised clinical trial designed to assess the effectiveness of pravastatin in reducing coronary mortality in individuals with a previous history of cardiovascular disease. The characteristics of individuals in the Aus-Diab cohort and the LIPID trial are summarised in Supplementary Table 1.

We measured 747 individual lipid species in the AusDiab cohort and 342 in the LIPID trial in 2018 and 2014, respectively, with the main difference between the assay methods being in the chromatographic conditions. The AusDiab study was analysed using IPA/ACN chromatography with a dual column system and the solvent consisting of isopropanol, acetonitrile and water, whereas the LIPID study utilised THF chromatography with a single column and the solvent based on tetrahydrofuran, methanol and water. In both studies, assay performance was monitored using plasma quality control (PQC) samples. However, PQC samples were from different batches, and in the Aus-Diab we also included additional reference samples from the National Institute of Standards and Technology (NIST, SRM 1950).

Lipid analysis for the SAFHS was performed under similar conditions as the AusDiab analysis (see Methods section). We obtained complete measurements of 795 individual lipid species in the SAFHS cohort, but for the validation purpose in this study we only focused on 700 lipid species that overlapped between the AusDiab and SAFHS cohorts.

### Nomenclature alignment of lipid species across the AusDiab and LIPID studies
The naïve method of aligning platforms involves matching each lipid species by name to the single best possible match in the reference data. However, as the chromatographic separation of lipid species has improved over the years, what was regarded as a single lipid in older analyses could be represented as a combination of multiple isomeric lipid species in later analyses. This implies that some lipid species in

the target data should be mapped to a composite measure (linear sum) of two or more lipid species in the reference data. Supplementary Data 1 provides mapping between all lipids in our target data (the LIPID trial) and either the single best possible match or the composite measures created for this purpose in the reference data (the AusDiab cohort). Overall, 225 lipid species in the LIPID uniquely mapped to a single lipid species in the AusDiab dataset. There were 70 species that each mapped to 2 lipid species in the AusDiab dataset, 12 species that each mapped to 3 lipid species in the AusDiab dataset and the remaining 35 lipid species did not map well to any lipid species measured in the AusDiab study. A total of 176 lipid species in the AusDiab reference dataset contributed to building 82 composite lipids that were mapped to the LIPID study. Together with 225 individually matching lipid species, this gave us a set of 307 lipid measures to build models for predicting 346 lipid species measured in the AusDiab, but not in the LIPID study, and to expand the 82 composite lipids into 176 lipid species as measured in the AusDiab study (Supplementary Data 2).

### Identification of discrepancies between aligned lipid species
When aligning lipid species between datasets, some lipid species displayed dissimilar variation due to methodological differences between platforms, different lipidome predictors or incorrect annotations. Such discrepancies in measurements or annotations can lead to local differences in the lipid correlation structure between the reference and target datasets, which in turn violates an important assumption for the successful prediction of lipid concentrations. To identify discordant lipid species and remove them prior to building predictive models, we used differences in the partial correlation coefficients of corresponding lipid species between the two datasets.

Partial correlation – or conditional correlation in the context of a multivariate normal set of random variables – represents the strength of linear association between two variables when the effect of the remaining variables in the set is removed or controlled for. It avoids spurious correlations often seen when Pearson correlation is used in the multivariable setting. We assess the concordance between a pair of corresponding lipids in different datasets by calculating total squared distance between their partial correlation vectors in the two datasets. Here, the partial correlation vector of a lipid is a vector of partial correlations of that lipid with all other lipids individually. A large distance between a pair of corresponding partial correlation vectors in the reference and target dataset indicates the discrepancy between the association patterns of the corresponding lipid in the two datasets. This, in turn, signals that such a lipid will not be a suitable predictor in the predictive models applied on both datasets.

We used the following heuristics when determining a discrepancy threshold that warrants the removal of a lipid from the predictor set. Focusing on lipids shared between the reference (AusDiab) and target (LIPID) data, we identified and provisionally removed the most discordant lipid between the two datasets, as judged by the distance between its partial correlation vectors. In a sequential process, the partial correlation distances were recalculated after each lipid species removal and the process of the provisional removal continued until all remaining lipid species had between-data distances smaller than 2 median absolute deviations (MAD) from the median distance. We then used graphical methods (Fig. 1) to identify an optimal threshold for removing discordant lipids by analysing the distribution of the partial correlation distances between corresponding lipids (Fig. 1a) and average, per lipid, distance between the partial correlation matrices (Fig. 1b) of the two datasets. The former is well suited for identifying, on an individual lipid level, when to stop the provisional removal process i.e., when the distribution of the partial correlation distances shows no apparent outliers. The latter can be used to detect, on a global level, when the average distance between the two datasets is flattening and further removal of discordant lipids is having

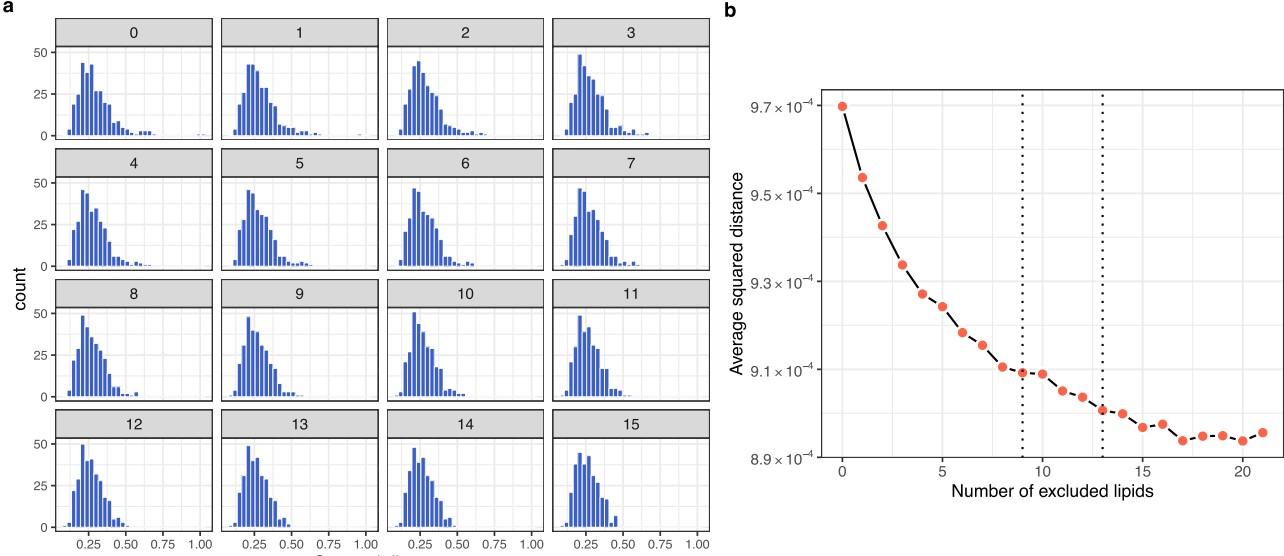

**Fig. 1 | Identifying the discordant lipid species between the AusDiab and LIPID studies. a** The distribution of the squared Euclidean distance between partial correlation vectors of corresponding lipid species in AusDiab and LIPID—a measure of discordance between identical lipid species in the two datasets. After each calculation, the most discordant lipid was removed from both datasets and partial correlation matrices as well as distances between corresponding lipids

recalculated. Facet labels indicate the number of lipids removed at the time. **b** The average squared distance between the remaining pairs of corresponding lipid species in AusDiab and LIPID after the removal of the most discordant lipid at each point. Dashed lines at 9 and 13 mark the points at which further removal of lipids would stop according to the rules of 3 and 2.5 MAD deviation from the median distance respectively. Source data are provided as a Source Data file.

diminishing return in terms of bringing the correlation structures of the two datasets closer together. In the depicted case, we used 2.5 MAD from the median distance as an optimal threshold for removing discordant lipids. This resulted in the removal of the 13 most discordant lipids from the predictor set (Supplementary Table 2).

Furthermore, partial correlation coefficients were useful for visualising association networks between lipids in the two datasets, comparing associations between lipid families and detecting any major discrepancies from the expected correlation structure. Figure 2 shows the lipid association networks for the AusDiab and the LIPID studies in which only the top 1000 associations were retained in each. The lipid associations that appear in the top 1000 in both datasets are highlighted as blue edges. Overall, this demonstrates very similar association patterns between lipid species and families in the two studies. In particular, we observed strong association between LPC and PC, LPE and PE, LPI and PI, DG and TG and between LPC and CE classes, the latter correlating specific fatty acid species across these classes. In addition, we used the Louvain community detection algorithm[10] to demonstrate that the modularity of lipid association networks largely overlapped with distinct lipid classes and created similar patterns of lipid class associations in the two datasets (Supplementary Fig. 1).

### Assessing transferability of predictive models from the reference to target study

To ensure successful transfer of predictive models from the reference (AusDiab) to target (LIPID) study, we first focused on the lipids measured in both datasets (matching lipids) as a form of validation. In this setting we can build models in the reference dataset, use them to predict the matching lipid concentrations in both the reference and target dataset, assess the prediction accuracy against observed, conceptually masked, lipid measurements and, most importantly, compare the accuracy of predictions between the reference and target dataset. Ideally, the prediction accuracy for the equivalent lipid species in the two datasets would be very similar, indicating a perfect transferability of predictive models from the reference to target study.

We employed elastic-net regression models ($\alpha = 0.1$) to predict individual lipid concentrations using the remaining matching lipids as

well as age, sex, BMI and lipid-lowering treatment as predictors. In addition, we examined the effect of removing discrepant lipid species from the predictor set on the prediction accuracy and transferability of models from the AusDiab reference to the LIPID study. The prediction accuracy was assessed as the strength of Pearson correlation between observed and predicted lipid concentrations. Figure 3a demonstrates that the removal of the most discrepant lipids, based on the partial correlation measure described above, also eliminated the most poorly predicted lipids in the LIPID study. Furthermore, a small improvement in the accuracy of predictions was achieved, relative to the reference AusDiab dataset, for a fraction of the remaining lipids (Fig. 3b). For most lipid species, the prediction accuracy in the LIPID study was comparable to the accuracy achieved in the AusDiab reference, where models were developed. This indicated that models built using the AusDiab study could be safely transferred to the LIPID study to predict the remaining unmeasured lipid species. Given the large sex discrepancy between the two datasets, we also investigated the effect of sex stratification on the prediction models. We observed no significant difference in the accuracy of mixed sex and sex-stratified predictive models (Supplementary Fig. 2), indicating that the correlation structure between and within lipid classes is sufficient to capture sex-dependent fluctuations in lipid concentrations.

Next, we demonstrated transferability and robustness of our imputation approach in situations where the two studies have significantly different clinical features, some of which could be highly predictive of the lipidome. To this end, we utilised the fact that participants in the AusDiab and LIPID studies are very different in their usage of lipid-lowering medications. While only 8.4% of AusDiab participants reported taking any lipid-lowering treatment, 50% of patients in the LIPID trail were taking pravastatin at the followup and 0% at the baseline (as a part of run-in phase). Knowing that pravastatin and similar lipid-lowering medications have significant and specific effects on lipidome, we wanted to compare the accuracy of imputations between placebo- and pravastatin-randomised patients at different timepoints. As above, we built predictive models for individual matching lipids in the AusDiab, one at the time, using the remaining matching lipids, and employed them to predict corresponding

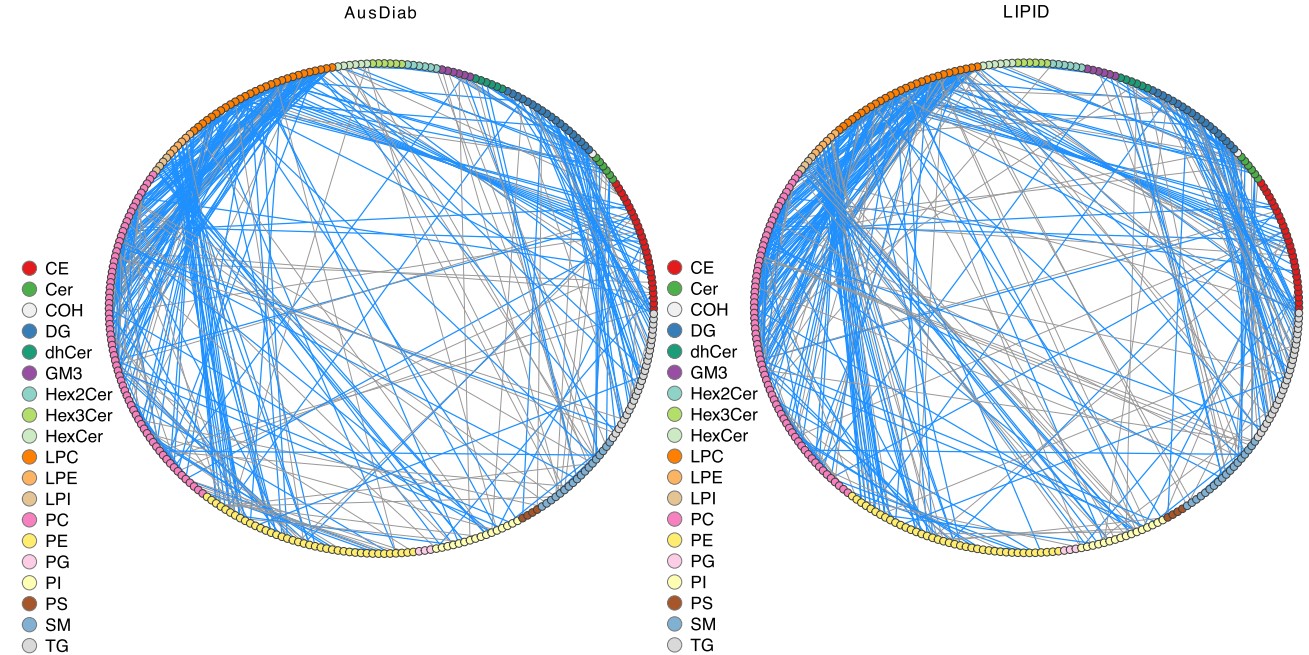

**Fig. 2 | Partial correlation networks between lipid species in the AusDiab and LIPID studies, presented in a circle layout.** Only lipids measured in both datasets were included in the analysis and the 1000 strongest associations are shown for each dataset. 683 edges highlighted in blue appear in top 1000 edges in both correlation networks. The legend shows a colour code for different lipid classes: CE cholesteryl ester, Cer ceramide, COH free cholesterol, DG diacylglycerol, dhCer dihydroceramide, GM3 GM3 ganglioside, HexCer monohexosylceramide, Hex2Cer dihexosylceramide, Hex3Cer trihexosylceramide, LPC lysophosphatidylcholine and lysoalkylphosphatidylcholine (LPC(O)), LPE lysophosphatidylethanolamine, LPI lysophosphatidylinositol, PC phosphatidylcholine, alkylphosphatidylcholine (PC(O)) and alkenylphosphatidylcholine (PC(P)), PE phosphatidylethanolamine, alkylphosphatidylethanolamine (PE(O)) and alkenylphosphatidylethanolamine (PE(P)), PG phosphatidylglycerol, PI phosphatidylinositol, PS phosphatidylserine, SM sphingomyelin, TG triacylglycerol. Source data are provided as a Source Data file.

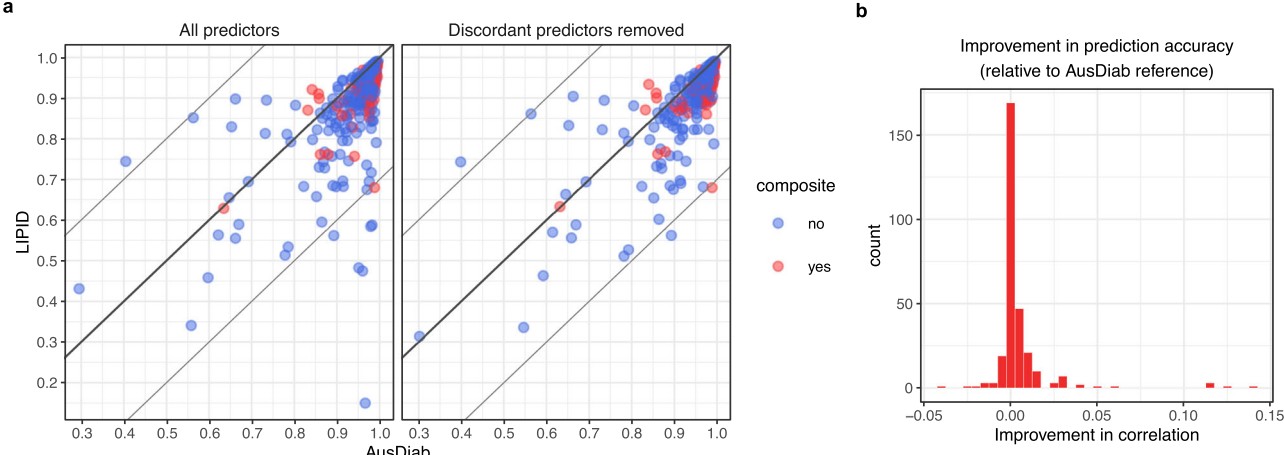

**Fig. 3 | Positive effect of removing discordant lipid species on the transferability of predictive models from the AusDiab to LIPID study. a** Scatter plots comparing the accuracy of lipid predictions in AusDiab and LIPID−before and after exclusion of 13 most discordant lipids from the models. Only lipids measured in both datasets were included in the analysis. Each dot represents prediction accuracy for a particular lipid species, assessed as the correlation between predicted and measured (conceptually masked) lipid concentrations within a given study. AusDiab predictions were assessed on a hold-out set; LIPID predictions were assessed on all observations. Individual lipid species (blue) and composite lipids (red). The line of an equivalent prediction accuracy in AusDiab and LIPID is shown; Thin lines mark the departure of 0.3, in correlation, from the equivalency line. **b** Improvement in the prediction accuracy relative to AusDiab: Absolute difference in accuracy of lipid predictions between AusDiab and LIPID after the removal of discordant lipids was subtracted from the corresponding absolute difference prior to the removal of any lipids. Source data are provided as a Source Data file.

individual lipids in the LIPID study. This time, the focus was widened to also include the follow-up timepoint, and accuracy of predictions was assessed within distinct subsets: placebo and pravastatin-randomised patients either at the baseline (when neither group is exposed to any lipid-lowering treatment) or at the followup (when only pravastatin group was exposed to treatment). Figure 4 demonstrates very similar accuracy of lipid predictions across these four groups, using two different views: focusing on the comparison of two randomisation groups at a particular timepoint (Fig. 4a, different patients, same timepoint), or of two timepoints for a particular randomisation group (Fig. 4b,

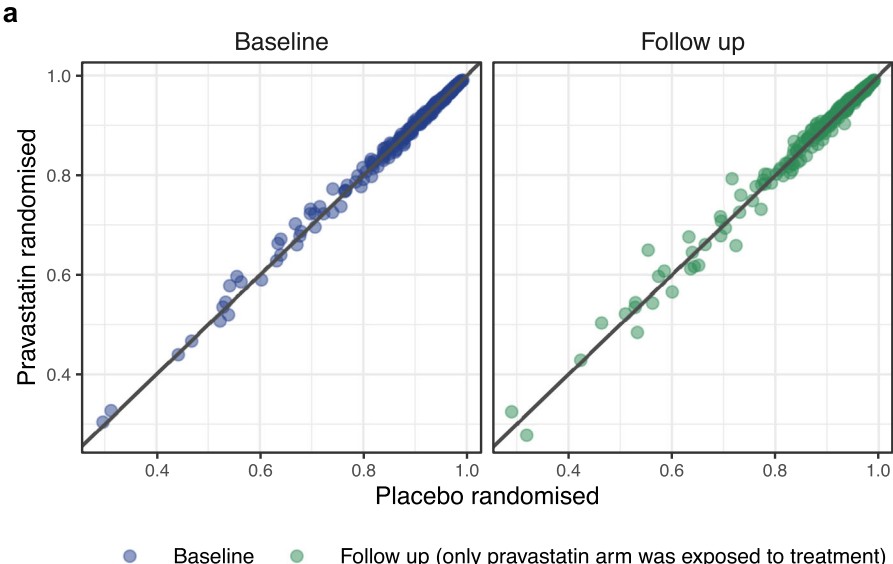

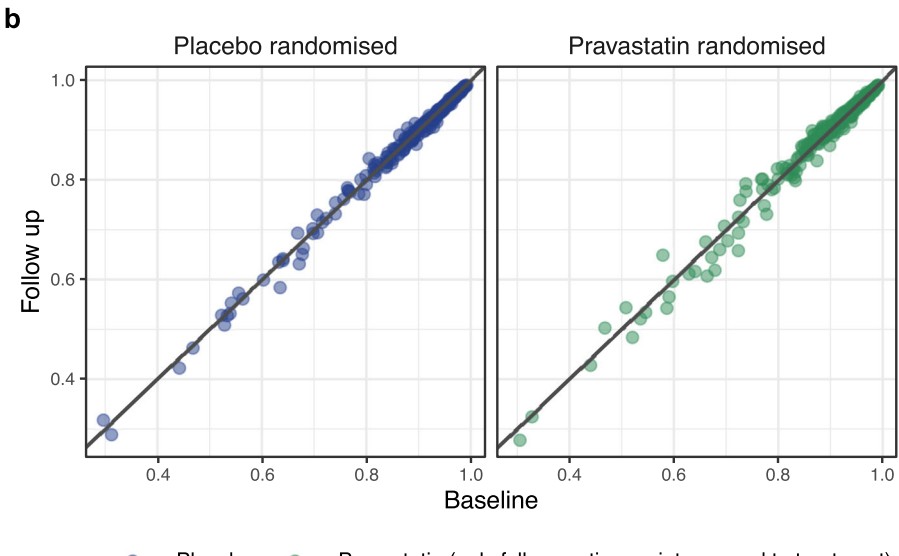

**Fig. 4 | Transferability of predictive models despite differential use of lipid-lowering treatments in the AusDiab and LIPID study. a** Scatter plots comparing the accuracy of lipid predictions in the LIPID study between placebo and pravastatin-randomised patient groups, at baseline (when neither group received treatment) and followup (when only pravastatin group was exposed to the treatment). Each dot represents prediction accuracy for one of 294 lipid species, assessed as the correlation between predicted and measured (conceptually masked) lipid concentrations within a given treatment group and timepoint subset.

Blue dots indicate no exposure to pravastatin in either compared subset, green dots indicate pravastatin exposure in one of the two subsets. The line marks an equivalent prediction accuracy in two given subsets. Only lipids measured in both datasets were included in the analysis, and 13 most discordant lipids were excluded from the models. **b** Comparing the accuracy of lipid predictions in the LIPID study between baseline and followup, in placebo randomised (neither timepoint received treatment) and pravastatin-randomised patient group (only follow-up subset was exposed to the treatment). Source data are provided as a Source Data file.

same patients, different timepoints). Furthermore, by repeating the same analysis after removing all non-lipid predictors from the models (age, sex, BMI, lipid-lowering medication), we achieved very similar results (Supplementary Fig. 3). This demonstrates predominant role of the lipid species and transferability of their correlation structure to achieve consistent accurate imputations across groups of cohorts with very different phenotypes.

After assessing transferability of the predictive models for individual lipids in terms of accuracy, we also wanted to assess the stability or precision of their predictions when the number of lipid predictors in the model was reduced. This was done to ensure that the removal of a larger number of discordant lipids, and therefore further reduction of the predictor set, in this or other studies, will not adversely affect the accuracy and precision of predictions. To this end, we compared

analyses in which 90, 75, 50 or 25 percent of the matching lipids were used to predict the remaining target lipid species. We randomly sampled individual targets and various reduced predictor sets to achieve 10 predictions of each target lipid. As above, we used Pearson correlation between observed and predicted concentrations for each lipid to measure prediction accuracy across resamples. Figure 5a shows the range of prediction accuracy for lipids in the AusDiab reference (blue) and the LIPID study (red), as the number of the predictor lipid species in models was reduced. When 10 to 25% of lipid predictors were excluded, we saw only a small decline in both the accuracy and precision of predictions in the LIPID study for most lipids. Figure 5b directly compares the average accuracy of lipid predictions in the AusDiab and LIPID studies, across all random predictor samplings. In a similar analysis, we trained models on progressively reduced sample

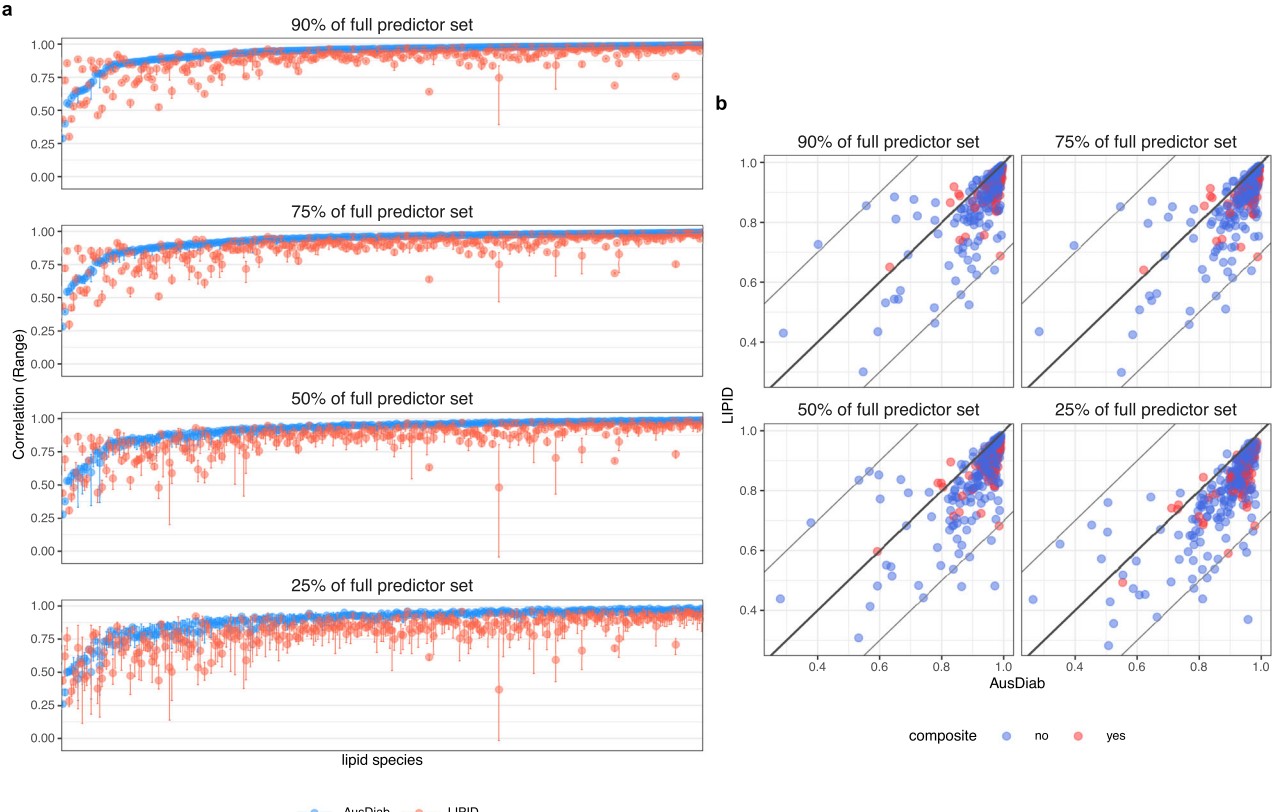

**Fig. 5 | Transferability of lipid prediction models from the Ausdiab to LIPID study - the effect of training models with reduced subsets of lipid species. a** The accuracy of lipid predictions, measured as the correlation between observed and predicted concentrations, in AusDiab (blue) and LIPID (red). The mean and range of correlations are shown, corresponding roughly to accuracy and precision, across $n = 10$ predictions of each individual lipid based on random subsamples of lipid predictors. Facet labels indicate the percentage of the initial lipid species retained in the predictor set when building models in AusDiab. 90, 75, 50 and 25% corresponds to 265, 220, 147 and 74 predictor lipids, respectively. Only lipids measured in both datasets were included in the analysis. AusDiab predictions were assessed on a hold-out set ($n = 1986$ observations); LIPID predictions were assessed on all baseline observations ($n = 5991$). **b** Scatter plots of the average accuracy of lipid predictions in AusDiab and LIPID, across 10 random subsamples of lipid predictors for each target lipid. Individual lipid species (blue) and composite lipids (red). The line of an equivalent prediction accuracy for a lipid in AusDiab and LIPID is drawn, as well as the departure of 0.3, in correlation. Source data are provided as a Source Data file.

size, instead of reduced predictor set (Supplementary Fig. 4a, b). The results indicated that 1000 or more observation was sufficient to build very stable models. Even the models built using only 200 observations at the time were surprisingly stable for most but not all lipid species.

Overall, the analysis demonstrated a good transferability of the elastic-net regression models for the prediction of lipid species. Although the LIPID trial is anthropometrically and clinically very different from our reference cohort (Supplementary Table 1), we showed a satisfying level of prediction accuracy and precision, when approximately 220 or more predictor lipid species and 1000 or more training observations were used to build predictive models.

### Assessing the prediction accuracy of missing lipids in the reference study

To examine how well the lipid species not measured in the LIPID study (the missing lipids) could be predicted, we first cross-validated their prediction in the reference AusDiab study, where the accuracy of predictions could be assessed against the observed lipid concentrations. Only if a lipid had been accurately predicted in the reference study, the corresponding model was used to predict the lipid in the LIPID study.

Here, the predictive model for an individual missing lipid was an elastic-net regression model that incorporated 294 concordant matching lipids as well as age, sex, BMI and lipid-lowering treatment as the predictors. We followed a 10-fold cross-validation approach when

assessing accuracy of prediction in the reference study, so that each lipid species was predicted once for every observation (individual). In addition to tuning the regularisation parameter $\lambda$ (determining the extent of regularisation), we also tuned the elastic-net parameter $\alpha$ (determining the type of regularised model) on a limited range (0, 0.1, 0.25, 0.5, 0.75) for each individual missing lipid. As the measure of prediction accuracy, we used the Pearson correlation between a full set of observed and predicted concentrations for each lipid.

Figure 6. shows very good prediction accuracy for the majority of lipid species. Lipids showing a correlation of less than 0.6 between predicted and observed values in the reference study were considered suboptimal prediction targets and were not predicted in the LIPID study. Supplementary Table 3 lists the 26 lipid species that were excluded from the prediction targets.

### Imputation and validation of imputed data

After excluding the 13 suboptimal predictors from the matching lipid species and the 26 suboptimal prediction targets from the missing lipids, we were able to predict 413 unmeasured lipid species in the LIPID study using 294 lipid species shared by the two datasets as well as age, sex, BMI and lipid-lowering treatment as predictors. Each unmeasured lipid was predicted using the elastic-net model with the elastic-net and regularisation penalties that achieved the best prediction accuracy for that lipid in the cross-validation analysis described above.

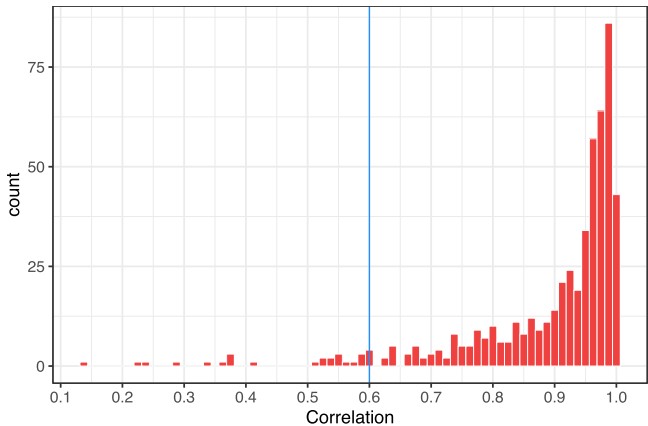

**Fig. 6 | Accuracy of predicting lipid species not measured in the LIPID study, within AusDiab data.** Histogram of correlations between observed and predicted lipid species. Glmnet model for each lipid was built using AusDiab data and included age, sex, BMI, lipid-lowering treatment and 294 compatible lipid species measured in both datasets as predictors. Predictions are assessed against 10 hold-out sets in a 10-fold cross-validation setting. Source data are provided as a Source Data file.

In addition to the single best prediction for each lipid, we also created 10 sets of stochastic predictions for each lipid (multiple imputations). This has been done in an effort to emulate the classic multiple imputation approach used for the imputation of data missing completely at random, or at random[11,12] and to compare validations of lipid species imputed in these two different ways. The aim was to confirm that no significant bias or underestimation of variability has been introduced in the downstream estimation or prediction procedures when using our single best prediction approach. Each of the multiple predictions were created with their own noise - normal noise with variance equal to the mean cross-validation error of the best-fitting model for that lipid. In this way, more uncertainty in the prediction was added to lipid species with less accurate models and higher cross-validation error.

To validate the predicted lipid species and compare them with the original lipid species in the LIPID study, we employed two types of assessment: (a) univariate associations of each lipid species with incident cardiovascular events; and (b) comparative assessment of the prediction accuracy for several anthropometric and clinical variables, when various sets of the original, imputed and multiple imputed lipid species were used as predictors. In the validations which included multiple imputed lipids, we performed analysis (estimation or prediction) with each of the 10 stochastic imputations separately, then pooled the results (parameter estimates or measures of predictive accuracy) according to the Rubin's rules[13,14].

Figure 7. shows univariate association of lipid species with the cardiovascular mortality over six years of followup for the lipid species originally measured in the LIPID trial and those imputed. Although they represent non-overlapping lipid species, the measured and imputed lipids reveal similar patterns of association with cardiovascular mortality across lipid classes, where they overlap. As expected, the pooled analysis of multiple imputed lipids showed similar point estimates to those obtained in the single best prediction analysis and similar but slightly smaller number of lipid species significantly associated with cardiovascular mortality. To provide further validation of our imputation approach, we performed the same association analysis focusing on the imputations of measured lipid species in the LIPID study (Fig. 8). In this setting, each individual measured lipid was conceptually masked and predicted using the remaining measured lipids as predictors. It was then possible to directly compare associations of the same set of measured and imputed lipid species with cardiovascular mortality and confirm their near-identity.

Next, we compared the ability of the measured and predicted lipids, as well as their combination, to predict several common variables such as cholesterol, systolic blood pressure (SBP), cardiovascular death and nonfatal stroke in the LIPID dataset (Table 1). In addition to lipids, each model also included age, sex and BMI in the predictor set. The prediction accuracy was similar when the measured or imputed lipid species were used to build prediction models, or when the combination of these two sets was used, despite different number of lipid species being used in these models. This indicated that no additional information was imposed onto predicted lipid species that could bias their associations with other, non-lipid variables. When comparing the models based on singly best imputed and multiple imputed lipids in terms of their average prediction accuracy and its standard error, there were instances when either the former or the latter were more similar to the models based on measured lipids. Therefore, no systematic bias or inflated confidence in the predictions was introduced under the models using singly best imputed lipids.

Finally, we employed our imputation approach to impute lipid species into another, ethnically distinct, target study the San Antonio Family Heart Study (SAFHS)[9,15]. Unlike the AusDiab, our reference study whose participants are predominantly of European origin, the SAFHS comprises solely of Mexican Americans participants. The two studies had 700 matching lipid species in common. This gave us an opportunity to conceptually mask a set of lipid species in the SAFHS similar to the set of missing lipid species in the LIPID trial, impute them using the models based on the remaining matching lipid species and objectively validate the accuracy of imputations by comparing observed (unmasked) and predicted lipid concentrations. Like in our earlier efforts, we first looked for any discrepancies between the sets of matching lipids to be used as predictors in the models and removed 12 discordant lipid species between AusDiab and SAFHS (Supplementary Fig. 5). We then focused on imputation of masked lipid species in SAFHS for which we could build sufficiently accurate predictive models in the AusDiab reference (lipid species showing correlation greater than 0.6 between their observed and predicted AusDiab concentrations). This gave us a set of 304 matching lipid species to build elastic-net regression models ($\alpha = 0.1$) in the AusDiab reference, and to predict concentrations of 384 masked lipid species, one at the time, in both the AusDiab and SAFHS studies. The true accuracy of predictions was assessed as the strength of correlation between observed and predicted lipid concentrations. Figure 9 demonstrates that, despite very different ethnic backgrounds, imputation models transferred seamlessly from the AusDiab reference to the Mexican American SAFHS cohort, achieving excellent accuracy of imputations for the majority of lipid species and strong linear relationship between imputation accuracies in the two studies.

## Discussion

Our aim was to design a strategy for harmonising lipidomic datasets produced using comparable lipidomic platforms that differ in their degree of isomeric separation and number of lipid species measured. This task is of a particular interest in association and prediction studies, when building inferential or predictive models and validating them across studies with different levels of granularity in their lipid measurements. Our approach of predicting (imputing) missing lipid species into a data with fewer measured lipids or lower degree of isomeric separation uses the regularised elastic-net regression and relies on the stable correlation structure, within and between lipid classes, across different lipidomic datasets.

To achieve the maximum possible isomeric separation in the imputed lipid species, equivalent to that found in the reference data, we looked for sets of lipid species in the reference data that best represent a single lipid in the less resolved target data. By establishing a mapping between these composite lipids (sum of two or more lipid species) in the reference data and their counterparts in the target data,

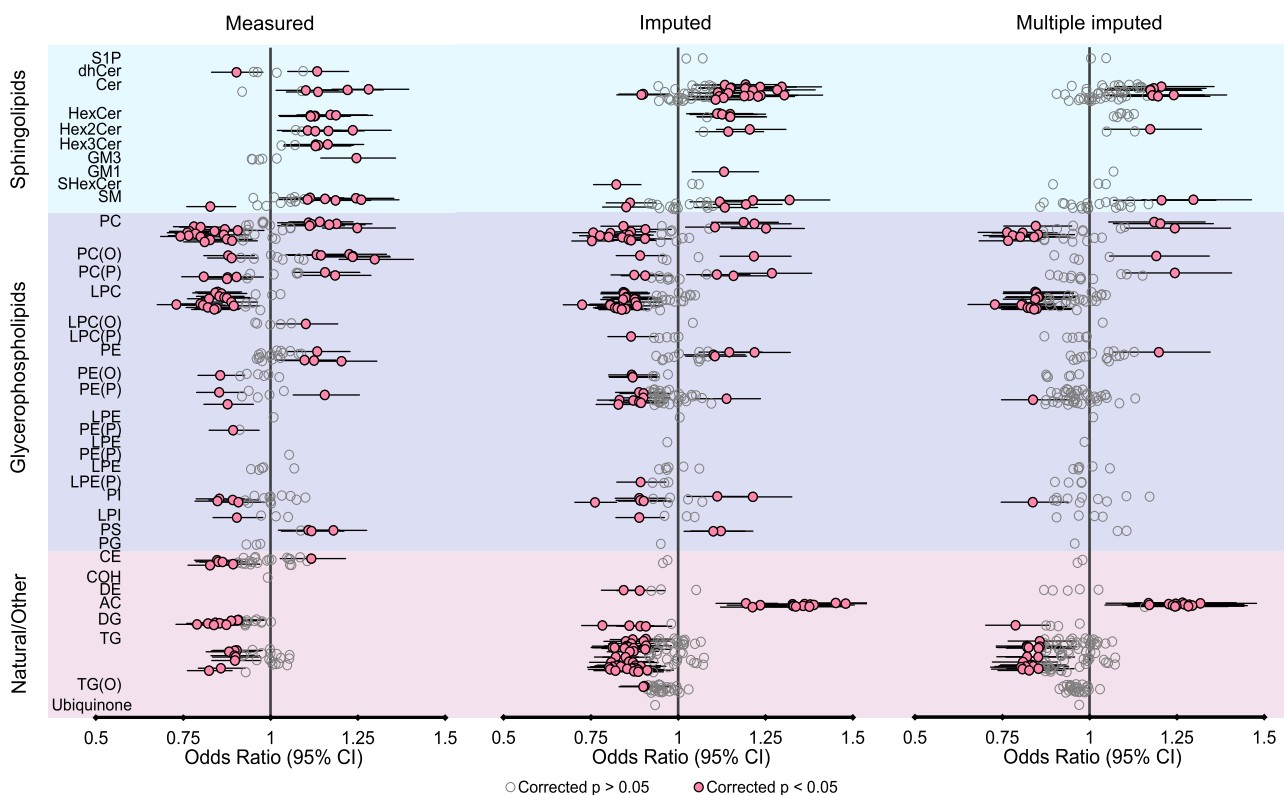

**Fig. 7 | Univariate association of measured and predicted missing lipid species with cardiovascular death outcome.** Logistic regression of cardiovascular mortality at 6 years followup against individual lipid species−either those measured in the LIPID trial at baseline, or the missing lipids imputed. Data are presented as odds ratio point estimates (dots) and 95% confidence intervals ($n = 5991$ observations). $P$ values were based on the Wald test, two-sided, and adjusted for multiple comparisons using Benjamini−Hochberg correction. Estimates for the multiple imputed lipids were derived by pooling 10 such logistic regression estimates according to Rubin's rules. AC acylcarnitine, CE cholesteryl ester, Cer ceramide, COH cholesterol, DE dehydrocholesterol, dhCer dihydroceramide, DG diacylglycerol, GM1 GM1 ganglioside, GM3 GM3 ganglioside, HexCer monohexosylceramide, Hex2Cer dihexosylceramide, Hex3Cer trihexosylceramide, LPC lysophosphatidylcholine, LPC(O) lysoalkylphosphatidylcholine, LPC(P) lysoalkenylphosphatidylcholine, LPE lysophosphatidylethanolamine, LPE(P) lysoalkenylphosphatidylethanolamine, LPI lysophosphatidylinositol, PC phosphatidylcholine, PC(O) alkylphosphatidylcholine, PC(P) alkenylphosphatidylcholine, PE phosphatidylethanolamine, PE(O) alkylphosphatidylethanolamine, PE(P) alkenylphosphatidylethanolamine, PG phosphatidylglycerol, PI phosphatidylinositol, PS phosphatidylserine, SHexCer sulfatide, S1P sphingosine 1 phosphate, SM sphingomyelin, TG triacylglycerol, TG(O) alkyl-diacylglycerol. Source data are provided as a Source Data file.

we improved the quality of predictors in our models. In addition, we were able to more accurately predict an expanded set of lipid species, which now included species that formed composite lipids in the reference data. In the case of the LIPID study, where we found 82 composite lipids corresponding to 176 lipid species in the AusDiab study, this led to a high-confidence prediction of additional 94 lipid species into the LIPID study. This approach requires careful annotation and mapping of lipid species and composite lipids. It may not be possible to map all lipids between datasets, however incorrect annotations should be identified and removed as done with discordant lipid species in this study.

Our prediction approach relies on the stable correlation structure within and between lipid classes, which is expected to be consistent between the reference and target data. To ensure this was the case within a subset of lipids common to both datasets (matching lipids) and that they can be used as reliable predictors in our models, we implemented a simple screening method based on partial correlations between matching lipids. Here, a measure of difference in the partial correlations of a lipid with other lipids between two datasets is taken as a measure of dissimilarity of the equivalent lipids between these two datasets. Based on this, we designed a dynamic and visual method for detecting and eliminating the most dissimilar lipid species, which would otherwise negatively affect our predictive models. We found that this approach performed better in identifying truly discordant pairs of corresponding lipids in two datasets than the similar approach

based on Pearson correlations. It also performed better than the approach focusing on the differences between multiple regression parameters in two datasets (data not shown) – likely due to symmetry and the scaled nature of the partial correlation coefficients.

Despite the considerable difference between the AusDiab, LIPID and SAFHS studies in terms of age, sex distribution, LDL/HDL ratio, ethnicity and cardiovascular health of the participants, we showed very similar lipid correlation structure between them and achieved similar levels of predictive accuracy for most lipid species that the studies have in common. Furthermore, we demonstrated robustness of our imputation approach even in situations when participants were very different in terms of their exposure to lipid-lowering medications. Despite specific effects of such treatments on the lipidome, we achieved very similar accuracy of lipid predictions in treated and non-treated groups of patients, and in the same group of patients before and after treatment. Thus, the correlation structure between and within lipid classes alone was sufficient to capture and reflect a significant heterogeneity of conditions affecting the lipidome between the two studies. Similarly, we have been able to replicate the success of our imputation approach into the ethnically distinct (Mexican American) SAFHS cohort. The importance of validating the accuracy of imputations with this additional study was two-fold. First, because the reference and SAFHS study had a large number of individually matching measured lipid species in common, it was possible to mask a similar set of lipids in the SAFHS as those missing in the LIPID study and

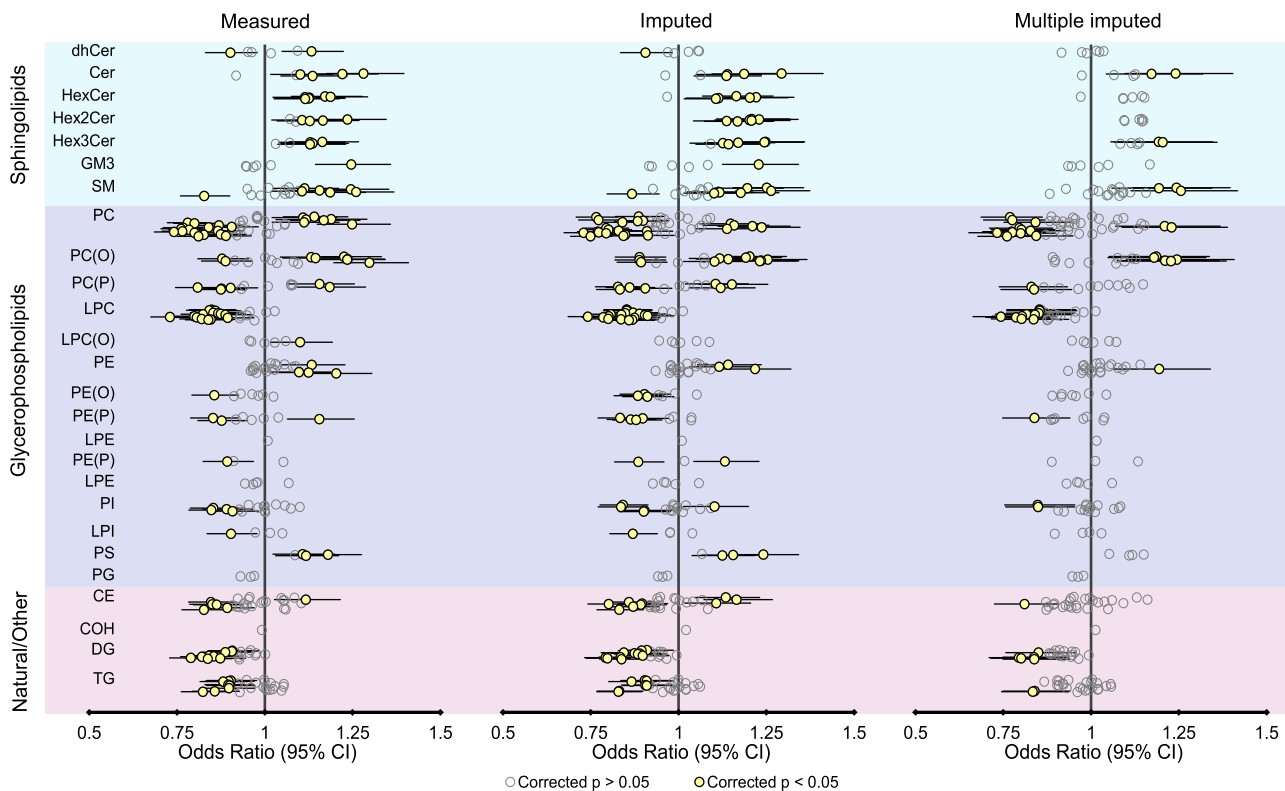

**Fig. 8 | Univariate association of truly measured lipid species and their predicted concentrations with cardiovascular death outcome.** Logistic regression of cardiovascular mortality at 6 years followup against individual lipid species−either those measured in the LIPID trial at baseline, or the same lipids imputed. Data are presented as odds ratio point estimates (dots) and 95% confidence intervals ($n = 5991$ observations). $P$ values were based on the Wald test, two-sided, and adjusted for multiple comparisons using Benjamini−Hochberg correction. Estimates for the multiple imputed lipids were derived by pooling 10 such logistic regression estimates according to Rubin's rules. CE cholesteryl ester, Cer ceramide, COH cholesterol, dhCer dihydroceramide, DG diacylglycerol, GM3 GM3 ganglioside, HexCer monohexosylceramide, Hex2Cer dihexosylceramide, Hex3Cer trihexosylceramide, LPC lysophosphatidylcholine, LPC(O) lysoalkylphosphatidylcholine, LPE lysophosphatidylethanolamine, LPI lysophosphatidylinositol, PC phosphatidylcholine, PC(O) alkylphosphatidylcholine, PC(P) alkenylphosphatidylcholine, PE phosphatidylethanolamine, PE(O) alkylphosphatidylethanolamine, PE(P) alkenylphosphatidylethanolamine, PG phosphatidylglycerol, PI phosphatidylinositol, PS phosphatidylserine, SM sphingomyelin, TG triacylglycerol. Source data are provided as a Source Data file.

perform the ultimate assessment of the accuracy of their imputations--comparison between measured and imputed concentrations. Second, it was possible to demonstrate transferability of such imputation methods between different ethnic groups. However, it is important to note that while some difference in the distribution of lipidomic predictors is tolerable, caution must be applied with the imputation of lipid species in situations where only one study harbours a condition or treatment that affects the lipidome in a way that disrupts the correlation structure. In such cases, it is important to remove any discordant lipid species and ensure the lipids correlation structure is comparable between the studies, as we have done in this study.

When assessing the predictive performance of our final models using cross-validation within the reference dataset (the AusDiab cohort), there were only 26 lipids which could not be predicted with the sufficient accuracy (correlation between the observed and predicted values greater than 0.6). Consequently, these lipid species were not predicted into the target dataset (the LIPID study). Some of the hard-to-predict lipid species came from lipid classes (Sph, S1P, SHexCer, Cer and DE) that were either absent or sparsely represented in the predictor set common to both the AusDiab and LIPID, which might explain the difficulty of predicting them accurately.

We provided an extensive validation of imputed lipid species by assessing their associations with a clinical outcome variable, cardiovascular mortality over 6 years of followup. It is difficult to compare the association of imputed and measured species with any outcome at the individual lipid level, as they are non-overlapping sets. However, we were able to confirm that the measured and imputed lipids exhibit the same direction of association with cardiovascular mortality within lipid classes (when they belong to the same class). In addition, we performed the same type of validation after conceptually masking and imputing the measured lipid species, one at the time. In this case, the associations between cardiovascular mortality and either imputed measured or truly measured lipids species were nearly indistinguishable. Furthermore, the associations of truly measured lipid species with cardiovascular mortality were more similar to those of singly imputed measured than to multiple imputed measured lipid species. This indicated that no concerning level of bias was introduced into estimates and no concerning underestimation of variability of estimates was introduced with our single best imputation approach. Indeed, we have been using large datasets and hundreds of lipid variables in our prediction models, giving us ability to create very realistic imputations.

Furthermore, we used a combination of measured and imputed or multiple imputed lipids to predict several numerical or categorical variables in the LIPID study and to compare the accuracy and precision of the predictions with those based solely on measured or imputed lipid species. The important outcome of such analysis was to verify that no significant departure in either accuracy or precision of predictions was noted when those different subsets of lipid variable were used. This confirmed that no additional information was imposed onto predicted lipid species that could bias their associations with other, non-lipid variables. Further research is needed to validate the use of

**Table 1 | Comparative ability of measured and predicted lipids alone or together to predict commonly reported numerical and categorical variables**

| Target variable | Data set | Accuracy[a] | 95% CI | SE |
|---|---|---|---|---|
| Systolic blood pressure | Measured | 0.28 | 0.256–0.303 | 0.0128 |
| | Imputed only | 0.284 | 0.268–0.299 | 0.0085 |
| | Measured + imputed | 0.285 | 0.257–0.312 | 0.0154 |
| | Multiple imputed only[b] | 0.277 | 0.255–0.298 | 0.0117 |
| | Measured + multiple imputed[b] | 0.277 | 0.26–0.294 | 0.0092 |
| Cholesterol | Measured | 0.826 | 0.818–0.833 | 0.0123 |
| | Imputed only | 0.822 | 0.811–0.834 | 0.018 |
| | Measured + imputed | 0.825 | 0.812–0.837 | 0.0204 |
| | Multiple imputed only[b] | 0.775 | 0.764–0.785 | 0.0132 |
| | Measured + multiple imputed[b] | 0.818 | 0.806–0.83 | 0.0186 |
| Cardiovascular death | Measured | 0.723 | 0.705–0.74 | 0.0088 |
| | Imputed only | 0.72 | 0.7–0.74 | 0.0102 |
| | Measured + imputed | 0.724 | 0.709–0.74 | 0.008 |
| | Multiple imputed only[b] | 0.7 | 0.681–0.719 | 0.0096 |
| | Measured + multiple imputed[b] | 0.71 | 0.691–0.73 | 0.0098 |
| Stroke | Measured | 0.529 | 0.503–0.554 | 0.013 |
| | Imputed only | 0.536 | 0.515–0.557 | 0.0108 |
| | Measured + imputed | 0.543 | 0.51–0.576 | 0.0169 |
| | Multiple imputed only[b] | 0.542 | 0.479–0.605 | 0.0321 |
| | Measured + multiple imputed[b] | 0.527 | 0.476–0.579 | 0.0262 |

Source data are provided as a Source Data file.

[a]Prediction accuracy was assessed as the correlation between observed and predicted values for the numerical variables (SBP and Cholesterol) and as the area under receiver operating curve (AUROC) for the categorical variables (Cardiovascular death and Stroke).

[b]Assessments based on multiple imputed data represent the average measure of prediction accuracy in which prediction accuracies from 10 stochastically imputed datasets were pooled using Rubin's rules commonly used in multiple imputation approaches.

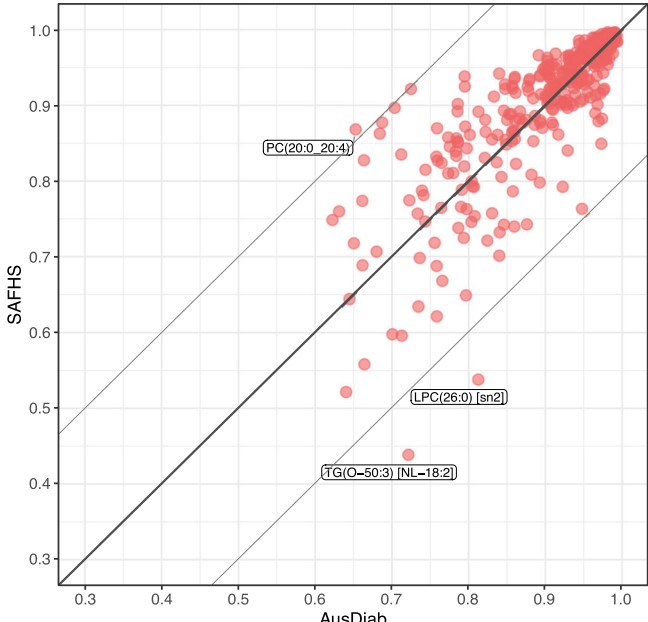

**Fig. 9 | Validation of lipid imputations in SAFHS—a Mexican American cohort, very different from the predominantly European descent AusDiab reference.** Scatter plots comparing the accuracy of lipid predictions in AusDiab and SAFHS. Each dot represents prediction accuracy for a particular lipid species, assessed as the correlation between predicted and measured (conceptually masked) lipid concentrations within a given study. Models were built using the AusDiab data to predict a similar set of lipids as those missing in the LIPID study for which accurate prediction models could be built in the reference AusDiab study (correlation between observed and predicted AusDiab concentrations above 0.6). AusDiab predictions were assessed on a hold-out set; SAFHS predictions were assessed on all observations. The line of an equivalent prediction accuracy in AusDiab and SAFHS is shown; Thin lines mark the departure of 0.2, in correlation, from the equivalency line. PC phosphatidylcholine, LPC lysophosphatidylcholine, NL neutral loss, sn stereospecific numbered (configuration of glycerol derivatives), TG triacylglycerol. Source data are provided as a Source Data file.

imputed lipid species in clinical risk prediction for the purpose of improved risk stratification.

These results clearly demonstrate the power of this approach to impute lipid species across datasets and so facilitate data sharing and integration. It is worth noting that plasma lipids may well be a best-case scenario because of the very strong correlation structure that existed both within and between lipid classes/subclasses. This is driven to a large extent by the interaction of the fatty acid metabolic pathways (lipogenesis, n3 and n6 fatty acid metabolism) and complex lipid pathways (phospholipid, sphingolipid, sterol and glycerolipid) whereby the fatty acid products from the synthesis pathways are incorporated as substrates into the complex lipid pathways creating these strong correlations both within and across lipid classes/subclasses that support the predictive models. Where such correlations do not exist, we see poorer performance of the models and so for many metabolites that do not share these common pathways we expect this approach to have limited utility. While the limitations of this imputation approach appear to be dependent on appropriate representation of the lipid classes of interest and a stable correlation structure across datasets, the ability to impute across lipidomic platforms of different modalities (i.e. reverse phase vs. HILIC vs. direct infusion) has yet to be formally tested and will require further analysis and validation.

Despite our success in accurately imputing lipid species across cohorts with different exposure to lipid-modifying treatments or different ethnicity, we would like to reiterate the notion that the best and most sensible imputation results are achieved when the reference and target studies are not very different with regards to relevant lipidome predictors. Importantly, all imputed lipids should be clearly reported/labelled as such (ideally with the estimated imputation accuracy) and a clear explanation of the imputation methodology.

In summary, we established a robust workflow for harmonising lipidomic datasets with different numbers of lipids species and different degrees of isomeric separation. This approach provides the opportunity to integrate plasma lipidomic datasets containing different lipid measures, acquired on similar platforms over the past decade or more. Such datasets can now be used to increase power for association analyses, which will be particularly important for GWAS where large datasets are required. The approach will also provide the opportunity for validation of predictive models across datasets even when some features of the models are absent from one dataset. As we move closer to clinical applications of lipidomic-based risk scores, our ability to integrate such population and clinical datasets will be a great advantage.

## Methods

This study used samples stored in the AusDiab and LIPID biobanks, which was approved by the Alfred Human Research Ethics Committee, AlfredHealth, Melbourne, Australia (project approval numbers,

AusDiab: 41/18, LIPID: 85/11 and 376/22). Studies were conducted in accordance with the ethical principles of the Declaration of Helsinki. No participant compensation was provided. Further validation was performed on the San Antonio Family Heart Study (SAFHS), which was reviewed and approved by the Institutional Review Board at the University of Texas Rio Grande Valley (IRB-18-0245, IRB-18-0255 and IRB-18-0406). The participants provided their written informed consent to participate in this study.

## Study populations

The AusDiab study started as a population-based cross-sectional survey of diabetes mellitus prevalence and associated conditions in the Australian adult population. This provided the baseline for a prospective cohort study aimed at identifying risk factors for diabetes mellitus and cardiovascular disease. The baseline survey was conducted between 1999 and 2000, with 11,247 participants aged 25 years and over drawn from 42 randomly selected urban and rural areas representing 7 states/territories of Australia, using a stratified cluster sampling method. A detailed description of the study population, sampling methods and response rates can be found elsewhere[6]. The measurement of anthropometric traits, behavioural risk factors and fasting serum clinical lipids was also reported[16]. Previously, we performed a comprehensive plasma lipidomic analysis on a total of 10,339 AusDiab study participants[3], as described below.

The LIPID study is a double-blind randomised clinical trial designed to assess the effectiveness of pravastatin in reducing coronary mortality in individuals with a history of cardiovascular disease (myocardial infarction or hospital admission for unstable angina pectoris). 9,014 patients between 31 and 75 years of age, who had total plasma cholesterol levels between 4 and 7 mmol/L and fasting triglycerides less than 5 mmol/L, were randomised to pravastatin (40 mg daily) or placebo. The detailed description of the trial, which recruited patients between 1990 and 1992 and had the median follow-up period of 6 years, can be found elsewhere[7,8]. Plasma samples and clinical lipid measurements were collected at baseline and the one-year followup. We performed a detailed plasma lipidomic analysis on 5991 participants for whom baseline and/or one-year fasting plasma samples were available[17,18], as described below.

The San Antonio Family Heart Study (SAFHS) is a prospective cohort study aimed at identifying genetic and environmental contributions to cardiovascular risk factors in Mexican Americans. The SAFHS enroled large, extended Mexican Americans families residing in San Antonio, TX, by way of the randomly selecting adult Mexican American probands, without regard to disease. The enrolment procedures, inclusion and exclusion criteria, and phenotypic assessments of the study participants have been previously described[9,19]. This is an ongoing investigation with several phases of data collection on 2595 individuals (qualifying first-, second- and third-degree relatives of the proband and proband's spouse as well as spouses of these relatives). We performed a comprehensive plasma lipidomic analysis on all SAFHS participants for whom fasting plasma samples were available at any data collection phase[15], as described below.

## Lipid extraction and liquid chromatography-mass spectrometry

**Lipid extraction.** The lipid extractions for the AusDiab, LIPID and SAFHS studies was carried out using a single-phase butanol/methanol method[20]. In brief, 10 μL of plasma was mixed with 100 μL of butanol/methanol (1:1) with 5 mM ammonium formate. A standard mix containing internal standards (AusDiab: Supplementary Data 3, LIPID: Supplementary Data 4) was included in the extraction solvent. Samples were vortexed thoroughly, followed by sonication for 60 min at room temperature. Each sample was subsequently centrifuged (16,000 × *g*, 10 min at room temperature) and the supernatant containing the lipid extract was collected and transferred into Teflon glass vials with 0.2 ml

glass inserts. The extracts were stored at −80 °C until analysed by liquid chromatography tandem mass spectrometry (LC-MS/MS).

**AusDiab lipidomics.** Lipid analysis for the AusDiab study was performed by liquid chromatography electrospray ionisation tandem mass spectrometry LC-ESI-MS/MS using an Agilent 6490 triple quadruple mass spectrometer (QQQ) with an Agilent 1290 series HPLC system and a ZORBAX eclipse plus C18 column (2.1 × 100 mm × 1.8 μm, Agilent). Mass spectrometry analysis was performed in a positive ion mode with dynamic scheduled multiple reaction monitoring (MRM) as detailed in the Supplementary Data 3. Chromatographic separation was carried out using a solvent system comprising of water: acetonitrile: isopropanol (Solvent A, 50:30:20, solvent B, 1:9:90% both with 10 mM ammonium formate) using a dual column setup. The column temperature was set to 45 °C with a flow rate of 0.4 mL/min.

Chromatographic conditions were as follows: starting at 15% solvent B and increasing to 50% B over 2.5 min, then quickly ramping to 57% B for 0.1 min. For 6.4 min, %B was increased to 70%, then increased to 93% over 0.1 min and increased to 96% over 1.9 min. The gradient was quickly ramped up to 100% B for 0.1 min and held at 100% B for a further 0.9 min. This was a total run time of 12 min. The column was then brought back down to 15% B for 0.2 min and held for another 0.7 min prior to switching to the alternate column for running the next sample. The column that was being equilibrated was run as follows: 0.9 min of 15% B, 0.1 min increase to 100% B and held for 5 min, decreasing back to 15% B over 0.1 min and held until it was switched for the next sample.

A 1 μL injection was used for each sample and the following mass spectrometer conditions were used: gas temperature, 150 °C; gas flow rate, 17 L/min; nebuliser, 20 psi; sheath gas temperature, 200 °C; capillary voltage, 3500 V and sheath gas flow, 10 L/min. Given the large sample size (n = 10,339), samples were run across several batches.

**LIPID lipidomics.** Lipid analysis for the LIPID study[17] was performed under similar conditions with the following differences: A shorter column was utilised (ZORBAX eclipse plus C18 column (2.1 × 50 mm × 1.8 μm)) with solvents A and B comprising of tetrahydrofuran: methanol: water in the ratio (20:20:60) and (75:20:5) respectively, both containing 10 mM ammonium formate. Column was heated to 40 °C and the auto-sampler regulated to 25 °C. Lipid species were separated under gradient conditions at a flow rate of 0.4 mL/min. The gradient was as follows: 0% solvent B to 40% solvent B over 2.0 min, 40% solvent B to 100% solvent B over 6.5 min, 0.5 min at 100% solvent B, a return to 0% solvent B over 0.5 min then 0.5 min at 0% solvent B prior to the next injection (total run time of 10 min per sample). Mass spectrometer source conditions were the same as the AusDiab lipidomic analysis. Given the large sample size (*n* = 5991 at baseline, *n* = 5782 at one-year followup), samples were run across several batches.

Supplementary Data 1 and 2 summarise all lipid species measured in the LIPID and AusDiab studies, respectively, as well as their mapping to the lipid species of the other study.

**SAFHS lipidomics.** Lipid analysis for the SAFHS study[15] was performed under a similar set of conditions to the AusDiab study, adjusted for a single-column setup. A ZORBAX eclipse plus C18 columns (2.1 × 100 × 1.8 μm, Agilent) was used heated to 45 °C. Mass spectrometry analysis was performed in both positive and negative ion mode with dynamic scheduled multiple reaction monitoring mode. The running solvent consisted of solvent A, 50% water: 30% acetonitrile: 20% isopropanol, containing 10 mM ammonium formate and 5 μM medronic acid, and solvent B, 1% water: 9% acetonitrile: 90% isopropanol, and containing 10 mM ammonium formate. The solvent conditions were: Starting at 15% solvent B, increased to 50% over 2.5 min, then to 57% over 0.1 min, then for the next 7.4 min increased to 70% B, 93% B over 0.1 min, 96% B over 1.9 min, then to 100% over

another 0.1 min and held at 100% B for 0.9 min for a total of 12 min. The solvent gradient was then switched back to 15% B over 0.2 min and held at 15% B for 3.8 min for a total run time of 16 min a sample.

The following mass spectrometer conditions were used: gas temperature 150 °C, gas flow rate 17 L/min, nebuliser 20 psi, sheath gas temperature 200 °C, sheath gas flow 10 L/min, capillary voltage and nozzle voltage of 3,500/1000 V (positive) and 3000 V/1500 V (negative). Isolation widths for Q1 and Q3 were set to unit resolution in both positive and negative mode (0.7 amu). Given the large sample size ($n = 5590$ samples derived from 2595 individuals), samples were run across several batches.

**Lipidomic data pre-processing**. To ensure the robustness of the lipid measures, we employed state-of-the-art lipidomic profiling techniques that are designed to capture a wide range of lipid species, including those with lower abundances. Integration of the chromatograms for the corresponding lipid species was performed using Agilent Mass Hunter, version 8.0 for the LIPID, and version 9.0 for the AusDiab and SAFHS studies. The relative concentration of lipid species was determined by comparing the peak areas of each lipid in each sample with the relevant internal standard. The up-to-date list of transitions and standards are available on https://metabolomics.baker.edu.au/. A median centring approach was carried out to correct for batch effect i.e. to remove technical batch variation using PQC samples[21]. Briefly, the lipidomic data for each species in each batch was aligned to the median value of that species for all the PQC samples that were included in each run. PQC samples consisting of a pooled plasma samples taken from healthy individuals and extracted alongside the study samples were incorporated into the analysis at 1 PQC per 20 study plasma samples. Technical quality control (TQC) samples were included in the runs every 20 samples to allow for the assessment of technical variation arising from the mass spectrometer. These were pooled samples that were extracted independently from the study samples and frozen in individual vials. The PQCs enable monitoring of the extraction and mass spectrometry run, while TQCs enable isolation of any issues to the mass spectrometer and were not used for any statistical analysis reported here. NIST 1950 reference plasma samples (Gaithersburg, MD, USA) were included for every 20 samples in the AusDiab and SAFHS studies (but not in the LIPID study) to facilitate future alignment with other studies.

## Predictive models and statistical analysis

All predictive modelling and analyses were performed using R (version 4.2.1)[22]. Elastic-net models were implemented in the `glmnet` package[23]. In all analyses, lipid concentrations were log-transformed, mean centred and scaled by the standard deviation. To predict missing lipid species in the LIPID study, we built models for each individual lipid by tuning the $\lambda$ and exploring a limited range of the $\alpha$ parameter (0, 0.1, 0.25, 0.5, 0.75) using a cross-validation framework within the AusDiab study. For the final prediction into the LIPID trial, we used the model with the $\alpha$ parameter value that achieved the best cross-validation predictive performance for that lipid and the largest $\lambda$ parameter value within one standard error of the minimum prediction error (`lambda.1se`). All other evaluative models were built by tuning only the $\lambda$ parameter while keeping $\alpha$ parameter constant at 0.1 and predicting outcomes with the $\lambda$ parameter values corresponding to the minimum cross-validation prediction error (`lambda.min`). For the evaluation of predictive performance of models, we used Pearson correlation between observed and predicted values for the numerical outcome variables and the area under receiver operating curve (AUROC, C-index) for the dichotomous outcome variables. The evaluations were based on the 10-fold cross-validation approach: each fold of data was predicted once, so that overall measure of predictive performance was based on predictions for all observations.

We performed two types of validation comparing the imputed lipid species against those originally measured in the target dataset. The estimation of univariate association between lipid species and cardiovascular mortality was performed with the measured, imputed and multiple imputed lipids using logistic regression. In addition, the evaluation of prediction accuracy for the selected variables was performed with the indicated combinations of measured, imputed and multiple imputed lipids using the elastic-net cross-validation. In validation analyses that included multiple imputed lipid species, either parameter estimation or evaluation of prediction accuracy was performed separately on each dataset, and subsequently pooled according to Rubin's rules[13,14]. The quality of the pooled estimates and the confidence intervals can be improved when pooling is performed in a scale for which the distribution is close to normal. Thus, we pooled parameter estimates from the logistic regression models on their original scale, before expressing the result in terms of odds ratios. Correlation coefficients were transformed using Fisher $z$ transformation[14] and the result back-transformed using the following formulas

$$z_i = \frac{1}{2}\ln\left(\frac{1+r_i}{1-r_i}\right); \bar{r} = \frac{e^{2\bar{z}}-1}{e^{2\bar{z}}+1} \tag{1}$$

AUROC/C-index were pooled on their original scale. The final point estimates and measures of prediction accuracy were the average of multiple estimates. The standard errors were computed using both the within and between imputation variance of the estimates. The within imputation variance was simply average variance of $m$ separate estimates $\hat{p}_i, i = 1,...,m$

$$v_W = \frac{1}{m}\sum_{i=1}^{m}\widehat{SE}_i^2 \tag{2}$$

where $\overline{SE}_i$ was either the standard error of parameter estimate for an imputed dataset $i$ or, in the case of the elastic-net predictive performance measure, derived from cross-validations within an imputed dataset $i$. Between imputation variance was the variance of the estimates across $m$ imputed datasets

$$v_B = \frac{1}{m-1}\sum_{i=1}^{m}\left(\hat{p}_i - \bar{p}\right)^2 \tag{3}$$

The total standard error of the estimate was then calculated as

$$SE_{\bar{p}} = \sqrt{v_W + v_B\left(1 + \frac{1}{m}\right)} \tag{4}$$

P values were adjusted for multiple comparisons (342 for measured and 413 for imputed lipid species) using the Benjamini–Hochberg correction[24].

## Reporting summary

Further information on research design is available in the Nature Portfolio Reporting Summary linked to this article.

## Data availability

Because of the participant consent obtained as part of the recruitment process for the AusDiab and LIPID studies, it is not possible to make data publicly available (including the individual deidentified data). Individual-level AusDiab data are available for analyses that do not conflict with ongoing studies, through application to the study lead Professor Jonathan Shaw and the AusDiab Study Committee (Email: Jonathan.Shaw@baker.edu.au). The timeframe for response to such requests is within two months. Individual-level LIPID data are available for analyses that do not conflict with

ongoing studies, through application to the study lead Professor John Simes and the LIPID Study Investigators (Email: John.Simes@sydney.edu.au). The timeframe for response to such requests is within two months. The SAFHS anthropometric and genomic data are publicly available through dbGaP (accession numbers: phs001215.v4.p2, phs000847.v2.p1, phs000462.v2.p1), whereas lipidomic data are available from J.E.C., (Email: joanne.curran@utrgv.edu) via a material transfer agreement for work consistent with the informed consent. The summary statistics for the AusDiab and LIPID studies are provided in the Supplementary files. Data generated in this study are provided in the Source Data file. Source data are provided with this paper.

## Code availability

Source code for this project is available on the GitHub repository (https://github.com/BakerMetabolomics/LIPID_imputation).

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

## Acknowledgements

This work was supported by The Heart Foundation 2020 Predictive Modelling Strategic Grant #105511; an award from the Ernest Heine Family Foundation; and the Victorian Government's Operational Infrastructure Support Program. The LIPID study was supported by Bristol-Myers Squibb and the NHMRC (grants 211086, 358395). The SAFHS was supported by R01 HL140681 and R01 DK127636 (for lipid data generation); and P01 HL045522, R01 MH078111, R01 MH078143 and R01 MH083824 (for SAFHS participant recruitment). K.H., D.J.M., J.S.S., and P.J.M. are supported by Investigator grants from the National Health and Medical Research Council of Australia. M.J.C. receives an endowed fellowship in the Cardiology Centre of Excellence from Filippo and Maria Casella. We would like to thank the investigators of the LIPID and Aus-Diab studies and the participants of these studies.

## Author contributions

Study conception and design: A.D., C.G., P.J.M. Data modelling, analysis, and interpretation: A.D. LC-MS/MS method development: C.G., K.H. LC-MS/MS performance, data processing and analysis: H.B.B., N.M, T.D. (T.D. also contributed to the design of Figs. 7 and 8). Statistical and analytic support: J.W., T.W., K.H., C.G. AusDiab data coordination: J.E.S., D.J.M. LIPID data coordination: J.S. SAFHS data coordination: J.E.C., J.B. Manuscript drafting and revisions: A.D., C.G., P.J.M. Data interpretation, manuscript review, and editing: A.D, C.G., P.J.M., K.H., D.J.M, J.E.S., M.J.C., M.I., J.Y.Y., G.A.F., J.S. P.J.M., and C.G. are the guarantors of this work and shall take the responsibility for the access to and integrity of data. All authors have approved the final version of the manuscript.

## Competing interests

The authors declare no competing interests.

## Additional information

[1]Baker Heart and Diabetes Institute, Melbourne, VIC 3004, Australia. [2]Baker Department of Cardiovascular Research, Translation and Implementation, La Trobe University, Melbourne, VIC 3086, Australia. [3]Baker Department of Cardiometabolic Health, The University of Melbourne, VIC 3010, Australia. [4]School of Mathematics and Statistics, The University of Sydney, Camperdown, NSW 2006, Australia. [5]Charles Perkins Centre, The University of Sydney, Camperdown, NSW 2006, Australia. [6]Kolling Institute of Medical Research, The University of Sydney, St Leonards, NSW 2065, Australia. [7]Department of Cardiology, Royal North Shore Hospital, St Leonards, NSW 2065, Australia. [8]Department of Human Genetics and South Texas Diabetes and Obesity Institute, School of Medicine at University of Texas Rio Grande Valley, Brownsville, TX, USA. [9]National Health and Medical Research Council of Australia (NHMRC) Clinical Trials Centre, University of Sydney, Sydney, NSW, Australia. [10]Department of Diabetes, Central Clinical School, Monash University, Clayton, VIC 3800, Australia. [11]These authors jointly supervised this work: Corey Giles, Peter J Meikle. ✉e-mail: corey.giles@baker.edu.au; peter.meikle@baker.edu.au

## LIPID Study Investigators

**John Simes**[9]

