## [Peer Review File · Nature Communications]

Reviewers' comments:

Reviewer #1 (Remarks to the Author):

In the manuscript by Dakic et al they demonstrate a framework for the imputation of lipids when two platforms have been used. This has an enormous possibility for researchers to go back into older datasets and impute lipids that were missed through the many excellent and well described reasons given in the manuscript. In general this is a well written manuscript presenting some important results. There are a few points that I think the authors should consider:

1) Given the difference in sex between the 2 cohorts and the fact that lipid metabolism in males and females are different did the authors consider splitting the data into sex and see if this makes any difference in the accuracy of the imputation?

2) I think the authors need to add a part in the discussion pointing out that this method will only work within similar modes of chromatography. For example I doubt very much that the imputation method would work if you were trying to impute reverse phase data from HPLC separation for example.

Reviewer #2 (Remarks to the Author):

Dakic et. al. present an imputation approach to recover the lipid species not detected in a cohort based on the lipidomic datasets in another (reference) cohort. The authors utilize elastic net to build prediction models and evaluate the accuracy by different types of analyses. The discordance between profiling platforms and the recovery of lipids in one cohort based on another are important topics in lipidomics. However, there are major concerns in this study and the manuscript. There are many improvements needed before being published. Please see my comments below.

1. A public tool or compiled source code is lacking for the proposed method.

2. The populations/cohorts used in this study are both large-scale and somewhat similar, i.e., collect baseline (pre-treatment) samples from participants at risk or having history of cardiovascular disease. The author did not discuss a key factor that may affect imputation performance: the heterogeneity between populations/cohorts. If researchers use a cohort of cancer patients/survivors to predict a cohort of non-cancer participants, the results may be biased.

3. It's hard to tell whether the removed discordant species are helpful or harmful for prediction, since sometimes a platform (cohort to be predicted) may be more robust than the other (reference cohort). The improvement of true vs predicted correlation shown in Fig 3 b. is not striking, since most of the difference values are around zero. A toy simulation is needed.

4. In the assessment of predicted models from the reference to target study (transferability), the authors randomly sampled the individual target lipids and the predictor sets of lipids. However, in the real datasets, the absence of lipid species/classes in an old profiling platform compared to new platform is NOT random and should depend on the limitation of old technology. It's necessary to simulate the missing lipid species by mimicking differences in technologies.

5. To evaluate univariate association, I suggest authors knock out (mask) some lipids in the predicted cohort and then compare odds ratios of true vs predicted for these masked lipids. The comparison between measured vs imputed (absent) lipids cannot validate the imputation.

6. Why is transferability only assessed with elastic net $\alpha=0.1$? I suggest considering different values of alphas.

7. The terminology of some statistical methods/models are not accurate and misleading, e.g., ridge regression is $\alpha=0$ instead of $\alpha=0.1$. I assume the method used in this research is elastic net with $0 < \alpha < 1$.

8. The authors only used correlation coefficient to evaluate performance. Another typical and robust evaluation metric for imputation accuracy is root mean squared errors (RMSE). It's necessary to include RMSE as well.

9. The English writing in this manuscript needs further polishing. Some sentences are confusing, e.g., page 12, "we found only 26 lipids which we could not predict with a correlation of less than 0.6 between the observed and predicted values".

Reviewer #3 (Remarks to the Author):

Overall:

The authors propose a method(s) for combining lipid datasets from different platforms for larger analysis post-hoc. I am wary of this given the poor quality of some lipidomics datasets, and the current issues with QA/QC even within an actual lipid dataset. Therefore, imputing new lipid concentrations and names from another dataset further makes a dataset, which is now partly imputed, of lower quality with many assumptions which are hard to tease apart. For example, the authors assume certain correlation across the lipidome which is preserved across different datasets, but these correlations can change depending on disease states, age, etc. and this is not thoroughly discussed. Hence for this to be worth publishing, I would like to see extensive validation and discussion of assumptions built into the imputed dataset. Specifically, taking the same samples even, running on a targeted and non-targeted platform, using a different dataset to predict the non-targeted data, and then using the actual non-targeted data to assess the accuracy of imputation. Currently validation is only done via statistical methods, and hence no experimental validation is done which is greatly needed.

Furthermore, the manuscript is vague throughout and it is hard to discern the purpose of the study and methods employed, especially in the abstract and introduction. No grand new findings are discovered using this approach, and what use cases would the authors propose? When does this methodology work and when would it not? Finally, how can other users reproduce this work for their own studies without a software or SOP? Something should be provided for users as otherwise this will likely not be used by the community.

The manuscript proposes methods to combine multiple lipidomics datasets acquired on different platforms for different studies, to perform larger studies using repositories. This is a noble endeavor indeed, because if lipidomics datasets could be combined from multiple studies with similar meta-data then new interesting questions due to the larger n could be achieved without reacquiring data. At first glance though, this seems impossible: Interlaboratory studies on lipid measurements show drastically different concentrations obtained from laboratory to laboratory, multiple issues of artifact lipids from sample preparation and instrument issues, and common cases of the incorrect annotation of lipids. In short it is currently challenging to come to consensus on lipid measurements across laboratories hence the quality of data and many artifacts makes it difficult to combine datasets.

Common cases of improper lipid annotation using high-resolution tandem mass spectrometry data and corresponding limitations in biological interpretation - PMC (nih.gov)

Optimization of Electrospray Ionization Source Parameters for Lipidomics To Reduce Misannotation of In-Source Fragments as Precursor Ions | Analytical Chemistry (acs.org)

Harmonizing lipidomics: NIST interlaboratory comparison exercise for lipidomics using SRM 1950-Metabolites in Frozen Human Plasma - PubMed (nih.gov)

Furthermore, when using different platforms and LC peaks may represent different types of lipids (e.g. one peak may represent 30 triglycerides, whereas with better separation many of these might form their own peak). The authors do mention this issue, but as mentioned above validation and assumptions must be discussed in depth.

The abstract is vague as far as what “imputation”, “resolution”, and “harmonization” mean since these can have many meanings. Clarify that resolution means mass resolution on the instrument, and/or when you mean structural resolution for annotation, or another meaning. Clarify what “target” and “reference” datasets are, maybe just use low and high mass spectral resolution dataset.

Also I only later realize the grand purpose of this work: to generate models off of a large number of lipidomics studies from different laboratories using different technologies, mention this purpose in the abstract.

“This means that older measurements can be composite measures of multiple lipid species that are entirely resolved in modern platforms”: and vice versa depending on LC conditions.

Line 86: “a large reference panel” of what? Clarify what you mean here.

Line 88: “we can build accurate predictive models for individual lipid species” predictive models of what? Concentration? Annotations? Across the same samples? Across different samples? Within a matrix? Vague.

Line 93: maybe introduce what the AusDiab cohort is and whether it is on the same patients as the LIPID trial and why these are being compared in the intro rather than results?

Line 133: I was confused at first because 225 lipid species mapping to a single lipid species I took literally as though these was one name is one dataset for 225 names in another dataset, and so on and so forth

for your other descriptions. Only after looking at the supplemental, I realized you meant: 225 lipid species uniquely matched in both datasets... clarify this sentence.

Line 146 "In some instances, due to methodological differences, variation and differences in nomenclature, we could not assume that lipid measurements contain the same information between platforms" Again this language is vague and more verbose than needed confusing the reader. What is "same information". Be more detailed and precise. For example: "When combining datasets, aligned features may actually represent different lipid species due to incorrect annotations. Furthermore, when combining lipid species from multiple features into one feature, trends across samples will be an average and may not be indicative of any one species." Or something of that nature...

The partial correlations analysis is very interesting, if I understand correctly is the assumption that for the same matrix (human blood) most lipid should correlate similarly with one another across individuals otherwise they are termed "discordant lipids". So if PC(16:0_18:1) goes up with LPC(18:1) it should go up across both datasets. Same thing for inverse correlations. Please clarify the assumptions and give an example like I did for this approach. I honestly don't think this is true because, for example, in some cases I might see PC(16:0_20:4) decrease as it releases FA(20:4) which increases across different levels of inflammation. Whereas in another dataset where inflammation is not a major condition, I see both increasing together, the more FA(20:4) the more PC(16:0_20:4) can be generated. What evidence do you have that the assumption of correlating structures should actually match across datasets and under what conditions is this valid?

Line 216: Again what does it mean to predict individual lipids? Concentration? Existence? Trends?

Validation and purpose: You show validation by generating predictive models based on all 3 datasets and seeing how well these predictions perform as well as including other metrics. I think validation by actually reanalyzing the same samples, or having some of the features blindly excluded and then imputed, and then matching the imputations to the actual values and seeing how well they line up is necessary... some actual experimental validation with follow up measurement would be ideal.

Note the excel files start on a blank cell to the far right (e.g. D2), might want to save them with A1 selected unless this is an issue with the Nature Communications system.

Response to reviewer's comments

Reviewer #1 (Remarks to the Author):

In the manuscript by Dakic et al they demonstrate a framework for the imputation of lipids when two platforms have been used. This has an enormous possibility for researchers to go back into older datasets and impute lipids that were missed through the many excellent and well described reasons given in the manuscript. In general this is a well written manuscript presenting some important results. There are a few points that I think the authors should consider:

1) Given the difference in sex between the 2 cohorts and the fact that lipid metabolism in males and females are different did the authors consider splitting the data into sex and see if this makes any difference in the accuracy of the imputation?

Response: The difference in sex is taken into account by including the sex variable in our predictive models. However, even when sex was excluded as a predictor, we did not notice any decrease in prediction accuracy. This suggests that the correlation structure between- and within-lipid classes is sufficient to capture sex-dependent fluctuations in lipid concentrations. Replicating the method and stratifying by sex effectively leads to half the sample size and doubling the number of predictors – substantially reducing power.

Nevertheless, we have now performed sex-stratified analysis and the plots below demonstrate no significant difference in the accuracy of mixed sex and sex-stratified predictive models. The analysis was focused on lipids measured in both AusDiab and LIPID study (matching lipids); Predictive models for individual lipids were build using the remaining matching lipids in AusDiab data; The models were used to predict corresponding individual lipids in the LIPID study and accuracy of their prediction was assessed as the correlation between predicted and measured ("masked") lipids in the LIPID study. Plots below compare the correlations obtained from such sex-stratified and mixed sex analyses (results of the latter being split into sexes to enable comparison). We now mention that sex stratification made no difference to the prediction models in the manuscript (**page 8, line 246**).

Figure. Sex stratified analysis of predictive models for lipid species.

2) I think the authors need to add a part in the discussion pointing out that this method will only work within similar modes of chromatography. For example I doubt very much that the imputation method would work if you were trying to impute reverse phase data from HILIC separation for example.

Response: Our method is designed to take advantage of correlations between quantitative measurements of lipid concentrations. Therefore, the type – or even presence – of chromatography is irrelevant, as long as the method outputs sufficiently accurate lipid concentrations. One consideration, however, is the overlap in lipid species that are captured by the different modes of separation. For accurate imputation, there needs to be a sufficient number of lipid species measured with the same resolution (whether intrinsically measured by the method or composite species created by summing measurements). We have expanded our discussion to address this information (page 13, line 430; page 16, line 516).

Reviewer #2 (Remarks to the Author):

Dacic et. al. present an imputation approach to recover the lipid species not detected in a cohort based on the lipidomic datasets in another (reference) cohort. The authors utilize elastic net to build prediction models and evaluate the accuracy by different types of analyses. The discordance between profiling platforms and the recovery of lipids in one cohort based on another are important topics in lipidomics. However, there are major concerns in this study and the manuscript. There are many improvements needed before being published. Please see my comments below.

1. A public tool or compiled source code is lacking for the proposed method.

Response: We have created a public github repository to host the code for this project (https://github.com/BakerMetabolomics/LIPID_imputation) (page 21, line 669). The code is well documented and allows anyone to replicate the analysis with their own data. We do note that due to data sharing limitations, we are unable to provide access to the individual level data for the studies used in this manuscript.

However, requests for this data can be made to the appropriate steering committees (see Data Availability Statement).

2. The populations/cohorts used in this study are both large-scale and somewhat similar, i.e., collect baseline (pre-treatment) samples from participants at risk or having history of cardiovascular disease. The author did not discuss a key factor that may affect imputation performance: the heterogeneity between populations/cohorts. If researchers use a cohort of cancer patients/survivors to predict a cohort of non-cancer participants, the results may be biased.

Response: The reviewer makes a good point here. However, we do not consider the two cohorts used in this study to be similar (summarised in Supplementary Table 1). The reference (AusDiab) is a general population cohort without specifically increased (average) risk of CVD at baseline. In contrast, the LIPID study is a clinical trial where all patients had experienced either myocardial infarction or unstable angina pectoris at enrolment, often accompanied with other co-morbidities and as such are under increased risk of secondary CVD. Furthermore, the LIPID study patients are, on average, more than 10 years older than AusDiab participants, with much higher LDL/HDL ratio and are predominantly males. We believe the "heterogeneity" in conditions across studies is not the obstacle in capturing lipid correlation structure (as demonstrated in this manuscript), as long as important determinants of the lipidome, such as a disease, is present in both datasets (though not necessarily equally present). We expanded our results section (**page 8, line 253**) and added Figure 4. and Supp. Figure 2, to further demonstrate robustness of our imputation approach to "heterogeneity" in lipid-lowering treatment across studies. Only about 8% of AusDiab participants used lipid-lowering medications, while 0% of LIPID patients at baseline and 50% patients at follow up used pravastatin. We showed very similar accuracy of lipid predictions between pre-treatment and post-treatment arms of the LIPID trial, despite known and significant effect of pravastatin on the lipidome. However, in situations when a disease, condition or treatment is present in only one study that can affect the lipidome in a broad or specific way, caution must be applied with any imputation approach. Importantly, we assess the stability of the correlation structure, and exclude any lipids that show large deviations from it. We also expand our discussion of these concepts and emphasize the importance of verifying the stable correlation structure between and within the lipid classes (**page 14, line 450**).

3. It's hard to tell whether the removed discordant species are helpful for harmful for prediction, since sometimes a platform (cohort to be predicted) may be more robust than the other (reference cohort). The improvement of true vs predicted correlation shown in Fig 3 b. is not striking, since most of the difference values are around zero. A toy simulation is needed.

Response: Indeed, the improvements in true vs predicted correlation shown in Figure 3b aren't striking (although, for five lipid species it improved by more than 0.1). We believe this is due to the care and effort taken to align these two platforms. In situations where the platform developers aren't able to contribute their expert knowledge, the identification and removal of discordant species becomes all the more important. Thus, we included this step in our workflow as an important safety measure. Whether a discordant species is in the reference dataset or the cohort to be predicted, including it causes misalignment of the correlation matrices and

therefore affects imputation accuracy. We have written a toy simulation in R that demonstrates this (https://github.com/BakerMetabolomics/LIPID_imputation): The plot below shows the improvement in correlation between true and predicted values when a single discordant variable is removed from a simulated dataset. The improvement in correlation of 0.146 is seen.

The relatively small effect on prediction accuracy (correlation) when we remove a few of the worst variables is likely due to the many good predictors in the dataset and is a function of the strong correlation matrix that exists between plasma lipid species. Thus, improvement will always be modest amongst non-removed variables.

4. In the assessment of predicted models from the reference to target study (transferability), the authors randomly sampled the individual target lipids and the predictor sets of lipids. However, in the real datasets, the absence of lipid species/classes in an old profiling platform compared to new platform is NOT random and should depend on the limitation of old technology. It's necessary to simulate the missing lipid species by mimicking differences in technologies.

Response: We apologise for confusion on this part. We indeed did model the “older profiling platform” when assessing the transferability. We restricted the predictive models to only use lipids measured in both cohorts, ensuring that we are mimicking these differences. We then explored the robustness of the predictive models by further reducing these lipids by randomly sampling a subset (90%, 75%, 50%, or 25%) of them. As shown in Figure 5a, many lipids are robustly predicted even with a substantial reduction in available lipids.

5. To evaluate univariate association, I suggest authors knock out (mask) some lipids in the predicted cohort and then compare odds ratios of true vs predicted for these masked lipids. The comparison between measured vs imputed (absent) lipids cannot validate the imputation.

Response: We appreciate this suggestion by the reviewer. Indeed, we had performed this analysis and the results were originally presented in Supplementary

Figure 3. We have amended our description for clarity and Supplementary Figure 3 is now moved to main Figure 8.

6. Why is transferability only assessed with elastic net $\alpha=0.1$? I suggest considering different values of alphas.

Response: We thank the reviewer for this comment. Our investigation involved investigating different alpha parameters. We found that changing alpha had surprisingly negligible empirical effect on accuracy of predictions. We found that the optimal alpha was 0.1 for a majority of lipid species. In cases when the optimal alpha was different, the gain in the observed-predicted correlation over an alpha of 0.1 was negligible. As such, we utilized a parsimonious decision to use a single alpha for all transferability models. The best-performing alphas were used only for the final prediction of truly missing lipids.

7. The terminology of some statistical methods/models are not accurate and misleading, e.g., ridge regression is $\alpha=0$ instead of $\alpha=0.1$. I assume the method used in this research is elastic net with $0 < \alpha < 1$.

Response: We apologize for our casual use of “ridge regression” over “elastic net with $\alpha=0.1$ ”. We have updated the manuscript to ensure accuracy of terms.

8. The authors only used correlation coefficient to evaluate performance. Another typical and robust evaluation metric for imputation accuracy is root mean squared errors (RMSE). It's necessary to include RMSE as well.

Response: We appreciate the reviewer's comment, however, in the simple case of comparing measured and predicted variables on a continuous scale, there is a direct relationship between RMSE and correlation (r):

$$RMSE = \sqrt{1 - r_{y_i \hat{y}_i}^2} SD_y$$

In addition, all lipid species in this study are unit variance scaled ($SD=1$; see methods section) so the above formula reduces to:

$$RMSE = \sqrt{1 - r^2}$$

Therefore, there is no additional benefit of including RMSE as an evaluation metric in our analysis.

9. The English writing in this manuscript needs further polishing. Some sentences are confusing, e.g., page 12, “we found only 26 lipids which we could not predict with a correlation of less than 0.6 between the observed and predicted values”.

Response: Thank you for highlighting the need for improved clarity in our manuscript. We have reviewed the entire document for clarity.

Reviewer #3 (Remarks to the Author):

Overall:

The authors propose a method(s) for combining lipid datasets from different platforms for larger analysis post-hoc. I am wary of this given the poor quality of some lipidomics datasets, and the current issues with QA/QC even within an actual lipid dataset. Therefore, imputing new lipid concentrations and names from another dataset further makes a dataset, which is now partly imputed, of lower quality with many assumptions which are hard to tease apart.

Response: We understand the reviewer's comment and are in agreement regarding the issues with many lipidomic datasets. However, our laboratory has been at the forefront of high-throughput lipidomics for many years. Our methodology is promoted by Agilent Technologies as the gold standard for targeted lipidomics (<https://www.agilent.com/cs/library/applications/an-plasma-lipidomics-6495-lc-ms-ms-5994-3747en-agilent.pdf>). Furthermore, we host an open/transparent and comprehensive resource detailing our methodology (<https://metabolomics.baker.edu.au/method/>), which goes far beyond what many other laboratories offer.

The key issue here is that the datasets must be correctly aligned, and the correlation structures must be similar to support the accurate imputation. We accept that this is not a trivial task and for some datasets will not be possible. However, the potential gains in terms of statistical power and cost savings will make this an important approach in many larger well annotated studies.

For example, the authors assume certain correlation across the lipidome which is preserved across different datasets, but these correlations can change depending on disease states, age, etc. and this is not thoroughly discussed. Hence for this to be worth publishing, I would like to see extensive validation and discussion of assumptions built into the imputed dataset.

Response: Although the two datasets used in this study were quite different in terms of participant's age, sex, statin use, and health status, there was a similar correlation structure between lipid measurements. In Figure 2 we show that the partial correlations between lipid species is remarkably similar between datasets. Furthermore, we performed additional analyses and expanded our results section (added Figure 4. and Supp. Figure 2, Results section, **page 8, line 253** to further demonstrate robustness of our imputation approach to "heterogeneity" between studies. Please also see our response to Reviewer 2, comment 2, also concerning the correlation structure. In addition, a significant portion of our manuscript involves the identification and removal of discordant lipids i.e. lipids that show different correlations between datasets.

Specifically, taking the same samples even, running on a targeted and non-targeted platform, using a different dataset to predict the non-targeted data, and then using the actual non-targeted data to assess the accuracy of imputation. Currently validation is only done via statistical methods, and hence no experimental validation is done which is greatly needed.

Response: With respect, we disagree with the reviewer's suggestion that no experimental validation is done. We demonstrate the accuracy of our imputation approach using real data (~10,000 samples from the AusDiab cohort and ~10,000 samples from the LIPID study, where we impute several hundred lipid species). This

involved a range of careful internal checks performed on the set of lipid species common to both datasets, termed "matching lipids" in the text (also Figure 3). We then "masked" each of these matching lipids, one at the time, and built predictive models for these lipids in reference dataset (AusDiab) using all the remaining matching lipids, and used this model to predict the masked lipid in both the reference (AusDiab) and target (LIPID) studies. We then assessed the accuracy of these predictions within each dataset i.e. correlation between predicted and observed (unmasked) lipid in both AusDiab and LIPID, and then compared the two accuracies i.e. correlations. By doing this we confirmed that the accuracy of predictions in AusDiab and LIPID study were very similar.

Because matching lipids represent more than 40% of the total number of lipids in our reference data (AusDiab), it is reasonable to expect that the remainder of lipids (missing in LIPID) will not vastly depart from this trend. Furthermore, we limited imputation to only lipids which could be accurately predicted in the AusDiab reference dataset (with correlation > 0.6 between observed and predicted values).

We appreciate that the suggestion of imputation from a non-targeted dataset has a number of attractions, however these datasets are often not well annotated and as such, will present a particular series of challenges. We are indeed looking into the possibility of imputation from untargeted data but this is beyond the scope of this study.

Finally, how can other users reproduce this work for their own studies without a software or SOP? Something should be provided for users as otherwise this will likely not be used by the community.

Response: The source code for all procedures described in this manuscript are now openly available at https://github.com/BakerMetabolomics/LIPID_imputation. This will be included in the footnotes of the revised manuscript.

The manuscript proposes methods to combine multiple lipidomics datasets acquired on different platforms for different studies, to perform larger studies using repositories. This is a noble endeavor indeed, because if lipidomics datasets could be combined from multiple studies with similar meta-data then new interesting questions due to the larger n could be achieved without reacquiring data. At first glance though, this seems impossible: Interlaboratory studies on lipid measurements show drastically different concentrations obtained from laboratory to laboratory, multiple issues of artifact lipids from sample preparation and instrument issues, and common cases of the incorrect annotation of lipids. In short it is currently challenging to come to consensus on lipid measurements across laboratories hence the quality of data and many artifacts makes it difficult to combine datasets.

Common cases of improper lipid annotation using high-resolution tandem mass spectrometry data and corresponding limitations in biological interpretation - PMC (nih.gov)

Optimization of Electrospray Ionization Source Parameters for Lipidomics To Reduce Misannotation of In-Source Fragments as Precursor Ions | Analytical Chemistry (acs.org)

Harmonizing lipidomics: NIST interlaboratory comparison exercise for lipidomics using SRM 1950-Metabolites in Frozen Human Plasma - PubMed (nih.gov)

Response: We agree with the reviewer that combining lipidomic datasets from different laboratories presents a particular series of challenges that will need to be met to apply this methodology. For laboratories with high quality and well annotated data this will be possible, while for some laboratories this may not be possible. Importantly, in this manuscript we describe a process to assess the suitability of harmonised data and identifying lipid species that are not harmonised (i.e. mis-annotation or mixed signals). We further provide a process to develop and test algorithms for the proposed imputation. The gains to be made from such efforts to harmonise and meta analyse large datasets are very large and will be worth the effort in many instances.

Furthermore, when using different platforms and LC peaks may represent different types of lipids (e.g. one peak may represent 30 triglycerides, whereas with better separation many of these might form their own peak). The authors do mention this issue, but as mentioned above validation and assumptions must be discussed in depth.

Response: The reviewer is correct that different platforms will have better/worse separation of chromatographic peaks. This is the reason why we have generated the “composite lipids” in the Ausdiab dataset i.e. a single chromatographic peak in the LIPID study is represented by multiple chromatographic peak in the Ausdiab study, due to improved chromatography. By summing these individual lipid species, we replicate the reduced separation *in silico*, increasing the overlap between platforms. We have now discussed this issue in greater detail in the manuscript (**page 13, line 430**).

The abstract is vague as far as what “imputation”, “resolution”, and “harmonization” mean since these can have many meanings. Clarify that resolution means mass resolution on the instrument, and/or when you mean structural resolution for annotation, or another meaning. Clarify what “target” and “reference” datasets are, maybe just use low and high mass spectral resolution dataset.

Response: We apologize for ambiguity in the abstract. We have clarified or replaced these terms where appropriate.

Line 93: maybe introduce what the AusDiab cohort is and whether it is on the same patients as the LIPID trial and why these are being compared in the intro rather than results?

Response: We apologize for any lack of clarity. We have ensured that these cohorts are adequately described in the manuscript (Results section, **page 4, line 109**, and Supplementary Table 1).

Line 133: I was confused at first because 225 lipid species mapping to a single lipid species I took literally as though these was one name is one dataset for 225 names in another dataset, and so on and so forth for your other descriptions. Only after looking at the supplemental, I realized you meant: 225 lipid species uniquely matched in both datasets... clarify this sentence.

Response: We agree this was confusing. We have now clarified this section (page 5, line 143).

Line 146 “In some instances, due to methodological differences, variation and differences in nomenclature, we could not assume that lipid measurements contain the same information between platforms” Again this language is vague and more verbose than needed confusing the reader. What is “same information”. Be more detailed and precise. For example: “When combining datasets, aligned features may actually represent different lipid species due to incorrect annotations. Furthermore, when combining lipid species from multiple features into one feature, trends across samples will be an average and may not be indicative of any one species.” Or something of that nature...

Response: We thank the reviewer for pointing out issues with clarity and for suggesting improvements. We have been revised this section (page 5, line 158).

The partial correlations analysis is very interesting, if I understand correctly is the assumption that for the same matrix (human blood) most lipid should correlate similarly with one another across individuals lotehrwise they are termed “discordant lipids”. So if PC(16:0_18:1) goes up with LPC(18:1) it should go up across both datasets. Samke thing for inverse correlations. Please clarify the assumptions and give an example like I did for this approach. I honestly don't think this is true because, for example, in some cases I might see PC(16:0_20:4) decrease as it releases FA(20:4) which increases across different levels of inflammation. Whereas in another dataset where inflammation is not a major condition, I see both increasing together, the more FA(20:4) the more PC(16:0_20:4) can be generated. What evidence do you have that the assumption of correlating structures should actually match across datasets and under what conditions is this valid?

Response: Lipid-lipid correlations are determined by intrinsic biological pathways. These pathways ensure a certain co-ordination in the regulation of lipid metabolism. This results in correlations between lipid species that are generally stable across diverse physiological conditions. However, lipid metabolism is dynamic, responding to numerous biological factors, including inflammation, disease states etc. When these biological factors are unbalanced between datasets, the marginal correlations between lipid species can appear to change. However, if these confounding factors were controlled for (statistically regressed out), we would see a restoration of the expected, intrinsic lipid-lipid correlations. Partial correlations provide a way to measure these correlations without controlling for unmeasured factors. Therefore, discordant lipid-lipid partial correlations likely reflect differences in lipids measured across platforms (or the presence of additional, unaccounted effects in one of the datasets).

Thus, if PC(x) "goes up" together with LPC(x) in one dataset, that does not mean it must always "go up" in another dataset. We calculate partial correlation of PC(x) with each of the 300+ individual matching lipids in one dataset and likewise in another dataset. Then we calculate the distance between the two resulting vectors of PC(x) partial correlations (the sum of squared differences of paired elements of the two vectors). When distance between the two partial correlation vectors is large (as defined in Figure 1 and the relevant results section) we deem the corresponding lipid

to be "discordant" between the two datasets. The lipid can achieve this "discordant" status either by having moderately different partial correlations with a large number of other lipids between the two datasets ("globally discordant"), or by having strikingly different partial correlations with a small number of other lipids between the two datasets ("locally discordant").

Further to this, we performed additional analyses and expanded our results section (**page 8, line 253**) and added Figure 4. and Supp. Figure 2, to further demonstrate the robustness of our imputation approach to "heterogeneity" between studies. Please also see our response to Reviewer 2, comment 2, concerning the correlation structure.

Line 216: Again what does it mean to predict individual lipids? Concentration? Existence? Trends?

Response: We have now clarified that our aim is to predict the concentration of lipid species.

Validation and purpose: You show validation by generating predictive models based on all 3 datasets and seeing how well these predictions perform as well as including other metrics. I think validation by actually reanalyzing the same samples, or having some of the features blindly excluded and then imputed, and then matching the imputations to the actual values and seeing how well they line up is necessary... some actual experimental validation with follow up measurement would be ideal.

Response: Indeed, the blinding of features followed by imputation is a robust approach to validate these models. We have performed such analyses, with results shown in Figures 3, 4, 5, Supplementary Figure 2 and 3, with additional information on association of predicted and measured ("blinded") lipids with CVD available in Figure 8. We have endeavoured to ensure this is now adequately described in our revised manuscript.

Note the excel files start on a blank cell to the far right (e.g. D2), might want to save them with A1 selected unless this is an issue with the Nature Communications system.

Response: We thank the reviewer for this suggestion. We have selected A1 and saved the tables. We hope that this improves usability for reviewers and readers.

REVIEWER COMMENTS

Reviewer #1 (Remarks to the Author):

The reviewers have addressed my concerns and this can be published.

Reviewer #2 (Remarks to the Author):

The authors addressed some of my comments by including the source code for the proposed prediction model and simulation results. There are still points not fully addressed in the revised manuscript.

1. The heterogeneity across reference and target cohort is not extensively considered, which can be more profound than the example cohorts (AusDiab vs LIPID) in this manuscript. For example, adult vs pediatric (or infant) cohorts; cohorts representing different race or ethnicity; cohort of cancer patients receiving chemotherapy or radiation therapy vs cohort of non-cancerous patients. It is important to know whether the performance varies by heterogeneity between cohorts and how users will select the reference cohort, given the broad audience of this journal. Adding another pair of example cohorts is necessary.

2. The goal of this study stated in introduction is to predict unmeasured lipid species in the target dataset. However, the authors actually tried to zoom-in the measured lipids in target dataset to subclasses with higher resolution instead of recovering the missing parent classes in target dataset. This should be clarified throughout the manuscript.

Reviewer #3 (Remarks to the Author):

All my queries were addressed. The explanation of their validation is helpful and their reference of their methods and validation of these is helpful as well. The authors released their tool for the public as requested.

Some disclaimer in the software download page and in the manuscript that very clearly states the potential dangers of imputing datasets for users of all kinds of technical backgrounds who might use this tool should be warranted. It is understood that discordant lipids are left out, which helps in this issue. I am still very worry of coming across imputed lipid datasets in the future, and therefore a discussion of how the tool should and should not be used would be helpful. Careful guidelines should be delineated to the user with all assumptions clearly stated and potential risks in interpretation.

Response to reviewer's comments

Reviewer #1 (Remarks to the Author):

The reviewers have addressed my concerns and this can be published.

Response: We thank the reviewer for the time and effort dedicated to the revision process.

Reviewer #2 (Remarks to the Author):

The authors addressed some of my comments by including the source code for the proposed prediction model and simulation results. There are still points not fully addressed in the revised manuscript.

1. The heterogeneity across reference and target cohort is not extensively considered, which can be more profound than the example cohorts (AusDiab vs LIPID) in this manuscript. For example, adult vs pediatric (or infant) cohorts; **cohorts representing different race or ethnicity**; cohort of cancer patients receiving chemotherapy or radiation therapy vs cohort of non-cancerous patients. It is important to know whether the performance varies by heterogeneity between cohorts and how users will select the reference cohort, given the broad audience of this journal. Adding another pair of example cohorts is necessary.

Response: To investigate whether the performance varies when imputing into a heterogenous population, we have included an additional cohort representing a different ethnicity. The San Antonio Family Heart Study (SAFHS; n=2,595 individuals with 5,590 lipidomic samples) comprises solely of Mexican Americans selected from extended families residing in San Antonio, TX. In contrast, our AusDiab reference represents a random sample of Australian population in 1999/2000, which was of predominantly European ancestry (Australian census in 2001 reported only 1635 individuals of Mexican descent, or 0.007% of the Australian population). We demonstrate excellent transferability of the imputation models from the AusDiab reference to the Mexican American SAFHS cohort. These results are presented in the new Figure 9 and Supplementary Figure 4, described on page 12, line 391, and discussed on page 15, line 469.

Despite our success in accurately imputing lipid species across heterogeneous cohorts, we reiterated our conservative advice that the best and most sensible imputation results could be guaranteed when the reference and target studies are not extremely different (page 17, line 539).

2. The goal of this study stated in introduction is to predict unmeasured lipid species in the target dataset. However, the authors actually tried to zoom-in the measured lipids in target dataset to sub-classes with higher resolution instead of recovering the

missing parent classes in target dataset. This should be clarified throughout the manuscript.

Response: Indeed, our main goal was to predict unmeasured lipid species in the target dataset. As demonstrated in the manuscript, we imputed 413 new lipid species in the LIPID dataset. This includes 75 lipid species which are from (10) lipid classes that were not originally measured in the LIPID dataset. Similarly, in SAFHS, we demonstrated the same procedure replicates in a cohort of different ethnicity.

We believe the reviewer is referring to our validation experiments (Assessing transferability of predictive models from the reference to target study; page 7, line 214), where we empirically evaluated the imputation accuracy of lipids which were measured in both Ausdiab and LIPID. For these experiments, it was necessary to use the measured lipids and “zoom-in” to those of higher resolution in order to get an accurate imputation accuracy. However, these experiments were used to establish the performance of the method, rather than encompassing the entire method itself.

With the addition of the SAFHS cohort, we have performed a direct validation of all imputed lipids (direct validation was possible as these lipids were measured in the SAFHS, but conceptually masked for the validation purpose). This is described on page 12, line 391.

Reviewer #3 (Remarks to the Author):

All my queries were addressed. The explanation of their validation is helpful and their reference of their methods and validation of these is helpful as well. The authors released their tool for the public as requested.

Some disclaimer in the software download page and in the manuscript that very clearly states the potential dangers of imputing datasets for users of all kinds of technical backgrounds who might use this tool should be warranted. It is understood that discordant lipids are left out, which helps in this issue. I am still very wary of coming across imputed lipid datasets in the future, and therefore a discussion of how the tool should and should not be used would be helpful. Careful guidelines should be delineated to the user with all assumptions clearly stated and potential risks in interpretation.

Response: We thank the reviewer for their comment. We recognize the importance of a clear and prominent disclaimer to ensure that users understand the limitations, assumptions, and potential risks of the tool. We have included discussion in the manuscript outlining our recommendations for use and reporting of such datasets.

We would like to note that other fields – such as genomics – have readily integrated imputed datasets. A majority of genome-wide association summary statistics contain more than 90% imputed variables. Leveraging the experience from these fields, we recommend reporting quality scores for each imputed variable, providing the ability to distinguish original and imputed variables, as well as high- and low-confidence imputation.